
# Ensemble of optimised machine learning algorithms for predicting surface soil moisture content at global scale

Qianqian Han[1], Yijian Zeng[1], Lijie Zhang[2], Calimanut-Ionut Cira[3], Egor Prikaziuk[1], Ting Duan[1], Chao Wang[4], Brigitta Szabó[5], Salvatore Manfreda[6], Ruodan Zhuang[6], Bob Su[1,7,*]

[1]Faculty of Geo-Information Science and Earth Observation (ITC), University of Twente, 7514 AE Enschede, The Netherlands
[2]Research Center Jülich, Institute of Bio and Geosciences: Agrosphere (IBG-3), 52428 Jülich, Germany
[3]Departamento de Ingeniería Topográfica y Cartográfica, E.T.S.I. en Topografía, Geodesia y Cartografía, Universidad Politécnica de Madrid, Campus Sur, A-3, Km 7, 28031 Madrid, Spain
[4]Department of Earth, Marine and Environmental Sciences, University of North Carolina, Chapel Hill, NC, USA
[5]Institute for Soil Sciences, Centre for Agricultural Research, 1022 Budapest, Hungary
[6]Department of Civil, Architectural and Environmental Engineering, University of Naples Federico II, 80125 Naples, Italy
[7]Key Laboratory of Subsurface Hydrology and Ecological Effect in Arid Region of Ministry of Education, School of Water and Environment, Chang'an University, Xi'an 710054, China

*Corresponding author: Bob Su (z.su@utwente.nl)*

**Abstract.** Accurate information on surface soil moisture (SSM) content at a global scale under different climatic conditions is important for hydrological and climatological applications. Machine learning (ML) based systematic integration of in-situ hydrological measurements, complex environmental and climate data and satellite observation facilitate to generate the best data products to monitor and analyse the exchanges of water, energy and carbon in the Earth system at a proper space-time 20 resolution. This study investigates the estimation of daily SSM using eight optimised ML algorithms and ten ensemble models (constructed via model bootstrap aggregating techniques and five-fold cross-validation). The algorithmic implementations were trained and tested using the international soil moisture network (ISMN) data collected from 1722 stations distributed across the World. The result showed that K-neighbours Regressor (KNR) performs best on "test_random" set, while Random Forest Regressor (RFR) performs best on "test_temporal" and "test_independent-stations". 25 Independent evaluation on novel stations across different climate zones was conducted. For the optimised ML algorithms, the median RMSEs were below 0.1 cm3/cm3. GradientBoosting (GB), Multi-layer Perceptron Regressor (MLPR), Stochastic Gradient Descent Regressor (SGDR), and Random Forest Regressor (RFR) achieved a median r score of 0.6 in twelve, eleven, nine and nine climate zones, respectively, out of fifteen climate zones. The performance of ensemble models improved significantly with the median value of RMSE below 0.075 cm3/cm3 for all climate zones . All voting regressors 30 achieved the r scores of above 0.6 in thirteen climate zones except BSh and BWh because of the sparse distribution of training stations. The metrical evaluation showed that ensemble models can improve the performance of single ML algorithms and achieve more stable results. Based on the results computed for three different test sets, the ensemble model with KNR, RFR and XB performed the best. Overall, our investigation shows that ensemble machine learning algorithms





have a greater capability for predicting SSM compared to the optimised, or base ML algorithms, and indicates their huge

potential applicability in estimating water cycle budgets, managing irrigation and predicting crop yields.

## 1 Introduction

Surface soil moisture (SSM) plays an essential role in exchanges of water, energy and carbon between land and the atmosphere (Green et al. 2019) and affects vegetation and soil health, as well as the prediction and management of drought and flood events (Manfreda et al. 2017; Rodríguez-Iturbe and Porporato 2007; Su et al. 2003; Watson et al. 2022). SSM is

considered a key element in the feedback mechanisms that influence weather patterns and precipitation (Lou et al. 2021). The amount of soil moisture is largely determined by local climate, vegetation, soil type, and human activities, including irrigation and land use (Entekhabi et al. 2010a). While traditional ground-based observations provide valuable information, they often have limited spatial and temporal coverage. Remote Sensing (RS) techniques can provide near real-time, spatially explicit soil moisture information over large areas at a lower cost, and are particularly useful in areas with challenging terrain

or large, densely populated regions where it is not feasible to obtain ground-based measurements.

Both microwave, and optical and thermal infrared RS techniques have been used to estimate soil moisture (Eroglu et al. 2019). Passive microwave remote sensing is the most promising technique for global monitoring of soil moisture due to the direct relationship between soil emissivity and soil water content. It offers an advantage in providing observations in all-weather conditions and penetrating the vegetation canopy. There are many microwave radiometers used for soil moisture

observation and many soil moisture products have been generated using these microwave radiometers, such as special sensor microwave/imager (SSM/I), Advanced Microwave Scanning Radiometer for Earth Observation system (AMSR-E), soil moisture and ocean salinity (SMOS) and Soil Moisture Active Passive (SMAP). However, the derived soil moisture products from passive microwave sensors are limited by their coarse spatial resolution (generally from 9 km to 40 km) that impede the applications for regional scale studies. To enable applications at local scale, many studies have focused on downscaling soil

moisture using high-resolution optical and thermal and radar data (Fang et al. 2022; Song et al. 2022). The applicability of these downscaling algorithms is influenced by the need for a large amount of high-resolution data, which are not widely available on global scale. To obtain high-resolution soil moisture measurements, one promising strategy is to combine remotely sensed land surface radiometric temperature data with vegetation indexes. Additionally, it is possible to derive soil moisture from SMOS and AMSR-E data by applying hydrologic data assimilation approaches (Baldwin et al. 2017; Portal et

al. 2020).

Recently, Machine Learning (ML) techniques have gained popularity in several fields including soil moisture estimation (Ali et al. 2015; Han et al. 2023b; Zhang et al. 2021; Zhuang et al. 2023), due to their ability in identifying patterns and relationships between soil moisture observations and related predictors that may not be immediately obvious to a human analyst (Hajdu et al. 2018; Mao et al. 2019). This allows a ML model to make more accurate predictions of soil moisture

based on remote sensing data. Another advantage of using ML techniques is that they can help to reduce the uncertainty in



soil moisture estimation by accounting for the complex interactions between the soil, vegetation, and other factors that can affect the observed soil moisture. ML algorithms are capable of learning these relationships from the training data and use them to make more accurate predictions. Many efforts have been put into improving soil moisture prediction in the community using ML techniques (Abowarda et al. 2021; Karthikeyan and Mishra 2021; Lee et al. 2022; Lei et al. 2022;

Sungmin and Orth 2021; Zhang et al. 2021). At point scale, Uthayakumar compared three ML algorithms in the laboratory using a Radar Sensor (Uthayakumar et al. 2022). On a regional scale, Adab Acharya and Senyurek compared different ML approaches over catchment areas or at larger regional scales (Acharya et al. 2021; Adab et al. 2020; Senyurek et al. 2020). Liu compared six ML algorithms in generating high-resolution SSM over four regions (Liu et al. 2020). However, the comparison of different ML algorithms on soil moisture estimation with training data distributed across the globe is still

missing and the selection of predictors remains an open question.

Here we aim to optimise the prediction of soil moisture with training data distributed across the globe by ensemble models constructed from different base ML algorithms and extensively study their performances in order to identify optimized combinations for predicting soil moisture. This study aims to (i) optimise and compare the performance of the different ML approaches in soil moisture estimation based on the identical training and testing datasets across the globe (e.g. training

speed, accuracy, robustness, etc.); (ii) justify the selection of appropriate predictors and their importance in the ML model, (iii) build ensemble models for predicting soil moisture and compared with the results achieved by the optimised machine learning algorithms obtained in objective (i).

## 2. Data

The International Soil Moisture Network (ISMN) maintains a global in-situ soil moisture database through international

cooperation (ISMN, 2023). As of July 2021, ISMN contains more than 2842 stations from 71 networks over different climatological conditions (Dorigo et al. 2021). However, ISMN does not restrict data providers in terms of delivery intervals, automation, and formatting, resulting in heterogeneous data before harmonisation by ISMN. This includes variations in units, depth, integration length, sampling intervals, and sensor positioning (both vertically and horizontally), among others. To overcome these variations, ISMN harmonises soil moisture data by applying an automated quality control system, which

includes considering the geophysical dynamic range (i.e. threshold) and the shape of soil moisture time series (e.g., outliers, breaks, etc.) (Dorigo et al. 2013).

In this study, we extracted in-situ SSM from ISMN from 1 January 2000 to 31 December 2018, filtered the NaN (Not a Number) values of different predictor variables and kept 1722 stations for further analysis (Figure 1). Land surface information was also incorporated to prepare training and testing sets. In Section 2.1, we provide a detailed description of

each individual source of data. Section 2.2 explains the pre-processing operation applied to the available data, while Section 2.3 presents the data splitting procedure. It is worth noting that the original ISMN data is not fully openly accessible, and




registration is required for further inquiry or downloading. It is to note that the original ISMN data is accessible after registration (ISMN, 2023).

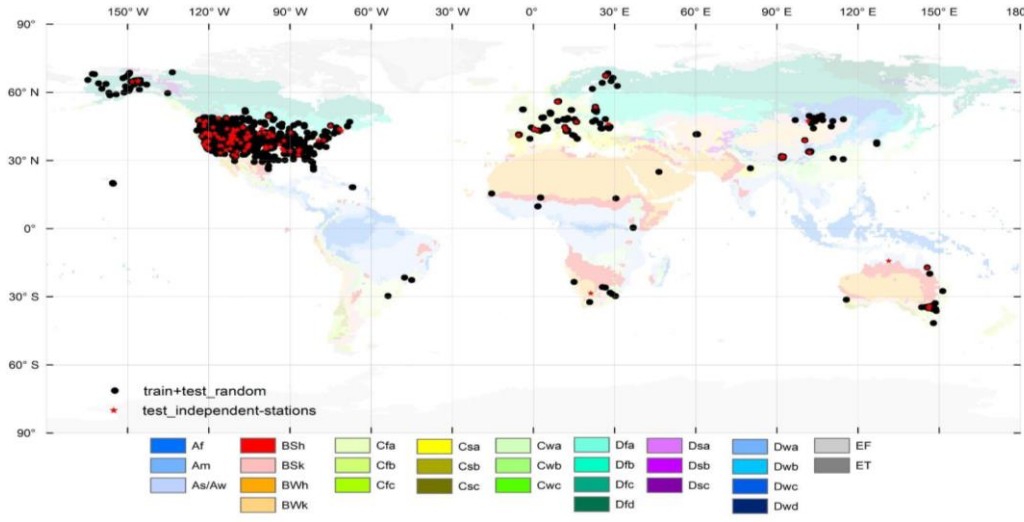

**Figure 1.** Spatial distribution of the ISMN stations considered in this study and their corresponding climate zones. Note: Data from the 1574 stations coloured in black was used for training and testing the performance of the algorithmic implementations, while the 148 stations coloured in red contain independent, unseen data from different locations (their data were used for post-training analysis).

### 2.1 Land Surface Features

The predictor variables used in the models are listed in Table 1 and detailed hereinafter. The most commonly used predictor variables (e.g. remote sensing data, or reanalysis data) are available on the Google Earth Engine platform.

**Table 1.** Predictor variables used for training the machine learning algorithms.

| Predictor Variable | Description/ Explanation | Spatial Resolution | Temporal Resolution | Source | Unit |
|---|---|---|---|---|---|
| Evaporation/ Precipitation/ API | Evaporation/ Precipitation/ Antecedent Precipitation Index (Weighted summation of daily precipitation amounts) | 11 km | Daily | ECMWF Reanalysis 5th Generation (ERA 5-Land) | Milimetre (mm) |
| NDVI/ EVI | Vegetation Index | 500 m | Daily | MOD13A1 | - |





| LST_Daily/ LST_Diff/ T_air | Daily land surface temperature/ Land surface temperature difference between day and night/ 2m air temperature | 1000 m/ 1000 m/ 0.25 degrees | Daily | MOD11A1/ MOD11A1/ ERA5-Land | Degree Celsius (℃) |
|---|---|---|---|---|---|
| Soil Texture/ Porosity/ Organic Matter Content (OMC) | Soil texture (proportion of clay, sand, silt)/ porosity: calculated from bulk density/ Soil organic matter content | 250 m | Static | SoilGrids | Percentage (%) |
| Lon/ Lat | Geographic coordinates information (Longitude and latitude) | - | Static | ISMN | Sexagesimal degree (°) |
| Elevation/ Topographic Index | Elevation/ topographic index | 90 m | Static | MERIT Hydro | metre (m) |
| Year/ DOY | Year/ Day of Year | - | - | - | - |

**2.1.1 Precipitation and Evaporation**

As the primary meteorological forcing, precipitation and evaporation control the spatial variability of SSM in most flat areas
(Pan et al. 2003; Wu et al. 2012; Zhang et al. 2019). Many studies have attempted to connect SSM with precipitation and evaporation, for example, with a linear stochastic partial differential model (Pan et al. 2003) and Antecedent Precipitation Index (API) (Shaw et al. 1997).

In this study, we used the hourly precipitation and evaporation data from ERA5-Land with a time coverage from 1981 to the present. ERA5-Land is one of the most advanced reanalysis products released by the European Centre for Medium-Range
Weather Forecasts (ECMWF), with higher spatial resolution and better global water balance than ERA-Interim (Albergel et al. 2018; Muñoz-Sabater et al. 2021). To synchronise the temporal coverages of the land surface temperature and Vegetation Indices, we adopted all precipitation and evaporation data from 2000 to 2019 and aggregated them from hourly into daily values.



### 2.1.2 Land Surface Temperature and Air Temperature

Land surface temperature (LST) reflects the pattern of evapotranspiration and plays an essential role in SSM retrieval (Parinussa et al. 2011). It was indicated that the GOES-8 satellite imagery derived LST increased with the decrease of observed SSM (Sun and Pinker 2004). Furthermore, the daily LST difference is negatively related to the thermal inertia of soil, while thermal inertia increases with soil moisture increase (Matsushima 2018; Paruta et al. 2020)(Zhuang et al. 2023). Thus, the daily difference between daytime and night-time LST was also selected as a predictor variable.

Currently, several LST datasets are available with rigorous validations. MOD11A1 (Collection 6) LST product from the Moderate Resolution Imaging Spectroradiometer (MODIS) is based on the split-window method (Wan 2014). The spatial resolution of the MOD11A1 is 1 km, with two measurements of LST per day (descending at local time 10:30, and ascending at 22:30, respectively). The MOD11A1 LST was reported with an average error of around one degree Celsius (Sobrino et al. 2020; Wan 2014).

### 2.1.3 Vegetation Indices


The vegetation index is a transformation of two or more spectral bands of satellite images. For example, the Normalised Difference Vegetation Index (NDVI) is one of the most used vegetation indexes, representing the greenness of the vegetation condition, and is considered as a conservative water stress index (Goward et al. 1991).

Plenty of research has been reported on retrieving SSM with the help of vegetation indices. For example, the
Temperature/Vegetation Dryness Index (TVDI) was shown to have a strong negative relationship with SSM (Patel et al. 2009). SSM was estimated with a random forest model using LST, albedo, and NDVI (Zhao et al. 2017). In addition, the Enhanced Vegetation Index (EVI) is also commonly used to improve the sensitivity of SSM estimation in areas with high vegetation coverage (Jiang et al. 2008; Matsushita et al. 2007).

This study deployed the MOD13A1 dataset of NDVI and EVI from MODIS as the predictor variables (MODIS 2015).
MOD13A1 has a spatial resolution of 500 m and the temporal resolution of 16-days. The selected temporal coverage is the same as for LST (from 1 January 2000 to 31 December 2018).

### 2.1.4 Soil Properties

In the case of soil moisture estimation, physics-based models are useful for predicting the movement of water in the soil based on physical factors such as temperature, precipitation, and soil physical properties (Sungmin and Orth 2021).
Nevertheless, the soil physical properties, such as sand, silt and clay content, organic matter content, were rarely included in empirical soil moisture models, although they can significantly influence the soil hydraulic processes (Vereecken et al. 2015). By considering these soil physical properties, empirical models can provide a better understanding of the mechanisms behind soil moisture dynamics, as they provide insight into the underlying processes that drive changes in soil moisture.





Soil properties influence the spatial variability of SSM, and among the most available ones, soil texture, porosity and organic
matter content (OMC) proved to play an important role (Van Looy et al. 2017). Soil texture refers to the fractions of clay, silt
and sand content. Porosity is the fraction of the total soil volume that is made up by the pore space, which varies depending
on other soil properties (e.g., soil texture, aggregation, etc.)(Lal and Shukla 2004). Soil organic matter is any material
originally produced by living organisms that is returned to the soil and goes through the decomposition process, and
represents an important soil component on a volume basis (Hudson 1994; Nath 2014).

Soil texture, organic matter content, and porosity determine the amount of water that can enter into the soil and be stored. In
this study, the soil texture (proportion of clay, sand and silt content), bulk density (used for calculating porosity), and organic
carbon content (used for calculating organic matter content) values were obtained from SoilGrids (Hengl et al. 2017). The
SoilGrids System provides currently the most detailed quantitative information on soil properties at global scale (Hengl et al.
2017). All soil properties are available for seven soil depths: 0, 5, 15, 30, 60, 100 and 200 cm, respectively (Hengl et al.
2017; Ross et al. 2018). In this study, clay, sand and silt content, organic matter content, and bulk density values of the top 5
cm were used.

### 2.1.5 DEM and Topographic Index (TI)

Topographic indices are often used to understand the soil moisture patterns in landscapes and make effective landscape
management decisions (Qiu et al. 2017). Digital elevation model (DEM) data were used because the distribution of
precipitation, vegetation, and other features, are directly related to elevation (Han et al. 2018). Topographic index (TI)
integrates the water supply from the upslope catchment area and the downslope water drainage for each cell in a DEM. In the
TI, the slope gradient approximates downslope water drainage, and the specific catchment area, calculated as the total
catchment area divided by the flow width, approximates the water supply from upslope area (Beven and Kirkby 1979).
In addition, the geographic coordinate of the soil moisture site was also added as a predictor variable. As discussed by
Zhang, the information of longitude denotes the closeness to the ocean, while the latitude is related to the climatology of the
temperature (Zhang et al. 2021).

### 2.1.6 Köppen–Geiger Climate Classification

To further investigate the performance of the ML algorithms over different climate conditions, we used the Köppen–Geiger
(KG) climate classification system. The KG system classifies the climate based on air temperature and precipitation. The
climate is grouped into five main classes with thirty sub-types, consisting of tropical, arid, temperate, continental, and polar
(Beck et al. 2018).
ISMN covers nineteen climate zones: Aw (Tropical wet and dry or savanna climate), BSh (Hot semi-arid climate), BSk
(Cold semi-arid climate), BWh (Hot desert climate), BWk (Cold desert climate), Cfa (Humid subtropical climate), Cfb
(Temperate oceanic climate), Csa (Hot-summer Mediterranean climate), Csb (Warm-summer Mediterranean climate), Cwb
(Subtropical highland climate), Dfa (Hot-summer humid continental climate), Dfb (Warm-summer humid continental



climate), Dfc (Subarctic climate), Dsb (Mediterranean-influenced warm-summer humid continental climate), Dsc (Mediterranean-influenced subarctic climate), Dwa (Monsoon-influenced hot-summer humid continental climate), Dwb (Monsoon-influenced warm-summer humid continental climate), Dwc (Monsoon-influenced subarctic climate), and ET (Tundra climate).

## 2.2 Data Pre-processing

### 2.2.1 Antecedent Precipitation Index

The ERA5-Land daily precipitation data was used to calculate the Antecedent Precipitation Index (API) (Muñoz-Sabater et al. 2021). API indicates the reverse-time-weighted summation of precipitation over a specified time (Wilke and McFarland 1986). The historical precipitation influences the soil water content in a weakening effect along the reverse time axis; the

more recent rainfall event has higher impact on the current SSM (Benkhaled et al. 2004).

Many researchers applied API to retrieve SSM information (Wilke and McFarland 1986; Zhao et al. 2011). In this study, we used the API as a feature for the SSM prediction. The definition of API can be represented with Eq. 1.

$$API_a = \sum_{i=0}^{t} k^i \cdot p_{a-i} \qquad (1)$$

In Eq. (1), $API_a$ represents the API value at day of $a$, $k$ is an empirical factor (decay parameter) to indicate the decay effect from the rainfall, which should always be less than one, a suggested range of k is between 0.85 and 0.98 (Ali et al.

2010), and $p_{a-i}$ is the precipitation value at $i^{th}$ day before day of $a$, and $t$ is the number of antecedent days we used to calculate $API_a$.

Despite the spatial heterogeneity of decay parameter (k), since the soil water retention varies from space, most researchers use only one pair of values (k and t) for their study area (Hillel and Hatfield 2005), which was adopted in this study as well. In this regards, we calculated the API with different combinations of the parameters (k and t) and compared the

corresponding Pearson Correlation Coefficient (r) of API and in-situ SSM, the finally obtained optimised parameters are k = 0.91 and t = 33.

### 2.2.2 Reconstruction of Vegetation Index

Both NDVI and EVI from MOD13A1 are MODIS 16-day composite data. Despite an atmospheric correction procedure for the MODIS reflectance data, noise could still be observed in the long-term time-series, which is not physical based on plant

phenology. Thus, we filtered the NDVI/EVI product with the Savitzky-Golay (S-G) method to reduce the small peak noise (Chen et al. 2004). NDVI/EVI were also interpolated to a daily time step by using a simple linear approach to synchronise the temporal steps with other features (applying Eq. 2).





$$p(t) = f(t0) + (f(t1) - f(t0))(\frac{t - t0}{t1 - t0}))$$ (2)

In Eq. 2, p(t) is the interpolated vegetation index, $f(t0)$ and $f(t1)$ are the vegetation index at time t0 and t1, respectively.

### 2.2.3 Daily LST and Daily LST Difference

The MOD11A1 LST product consists of two LST values per day (at 10:30 and 22:30 local time) (Wan 2014). We considered the arithmetic average of them as daily LST and calculated the difference between the daytime and night-time value as the Daily LST difference for that day.

The quality of the LST was ensured based on the quality control (QC) data associated with the daytime and night-time LST, only the pixels with the QC value of 0 (i.e., good quality data) were kept (Wan 2014). The MOD11A1 data used in this study

spans from 2000-02-24 to 2018-12-31.

### 2.2.4 Porosity and Organic Matter Content

Soil porosity was derived from Eq. 3, using the bulk density from SoilGrids (Hengl et al. 2017) and particle density.

$$\emptyset = 1 - \frac{\rho_b}{\rho_s}$$ (3)

In Eq. 3, $\rho_b$ is the dry bulk density (g cm$^{-3}$), and $\rho_s$ is the mineral particle density of 2.65 g cm$^{-3}$. For soil mixture, the bulk density scheme assumed that the coarse and fine components share the same particle density.

Soil organic matter content (also retrieved from SoilGrids) can be converted from soil organic carbon content by multiplying a factor of 1.72 (Khatoon et al. 2017).

### 2.2.5 DEM and Topographic Index (TI)

We used the topographic index of TOPMODEL (Kirkby 1975), which is defined in Eqs 4-6 by Pradhan as follows (Pradhan et al. 2006).

$$TI = ln(\frac{\alpha}{\beta})$$ (4)

$$\alpha = \frac{uca}{fw},$$ (5)



$$\text{fw} = \begin{cases} 90m, when\ flow\ direction\ is\ 1\ or\ 4\ or\ 16\ or\ 64, \\ \sqrt{90}\ m, when\ flow\ direction\ is\ 2\ or\ 8\ or\ 32\ or\ 128 \end{cases} \qquad (6)$$

In Eqs. 4-6, *TI* is the topographic index, $\alpha$ is the ratio of local upslope catchment area, *uca*, and flow width, *fw,* and $\beta$ is the slope angle of the ground surface that can be obtained from elevation data. Upslope catchment area, flow direction, and elevation can be found and used directly from MERIT Hydro: global hydrography datasets (Gruber and Peckham 2009; Yamazaki et al. 2019).

### 2.2.6 Spatial Resampling

Land surface features have different spatial resolutions. For calculating the long-term gridded global SSM, the predictor variables were resampled to 1-km resolution. Afterwards, the predictor variables were extracted for pixels that collocate with the considered in-situ sites, at 1 km resolution. The World Geodetic System of 1984 (WGS84, EPSG:4326) was chosen as the geographic coordinate system in our study.

### 2.3 Data Split

The precipitation, API, evaporation, air temperature, daily LST, daily LST difference, and NDVI/EVI data (described in Section 2.2) were synchronised based on the temporal coverage of in-situ data time-series of each ISMN station (described in Section 2.1). The full data contained a total of 735,475 registries (each sample contained nineteen predictor variables) and was differentiated in the training set and three different test sets, based on the following strategy (also described in Table 2):

First, we extracted 10% stations from every network for an independent evaluation of the derived ML models. This is called the "test_independent-stations" set and contains a total of 148 stations. The data from these 10% of the stations do not belong to either the training, or any testing set.

Afterwards, based on the temporal component of the data, we divided the remaining data (90% of the full available data) into "train&test_random" (70%; at this stage train&test_random form a single temporary set), and test_temporal set (containing 30% of the available data). This way, assuming the data were recorded from 1 January 2000 to 31 December 2019, the "train&test_random" set contains the first 70% data (14 years, from 2000 to 2013), and the "test_temporal" set consists of the last 30% data (6 years, from 2014 to 2019). The "test_temporal" set was used to analyse the performance of time-series prediction for stations where previous SSM data was used to build the ML model.

Finally, the temporary "train&test_random" data, obtained in the second step, was randomly split into the "train" and "test_random" sets by applying a 75:25% division criteria. The two resulting sets were used for training and testing the considered algorithmic implementations.

**Table 2.** Description of the sets of data used for training and testing the machine learning implementations and their size.

| ID | Name of the set | Percentage of total | Number of | Use description |
|----|----------------|---------------------|-----------|-----------------|
|    |                |                     |           |                 |





|  |  | data | samples |  |
|---|---|---|---|---|
| 1 | test_independent-stations | 10% | 66,155 | used for evaluating the performance of the models on unseen data belonging to new stations evenly spread at global level in space |
| 2 | test_temporal | 90% × 30% = 27% | 199,886 | used for computing and comparing the performance metrics achieved by the trained models on unseen data divided by temporal criteria |
| 3 | test_random | 90% × 70% × 25% = 15.75% | 117,359 | used for computing and comparing the performance metrics achieved by the trained models on unseen data divided by applying the randomisation criteria |
| 4 | train | 90% × 70% × 75% = 47.25% | 352,075 | used for training the model, cross-validation, and optimising the algorithms by applying hyper-parameters tuning |

## 3. Methodology

The optimization task of the considered ML algorithms involved an extensive search for the training hyperparameters that
achieved the highest performance metrics in different training scenarios (as illustrated in Figure 2).

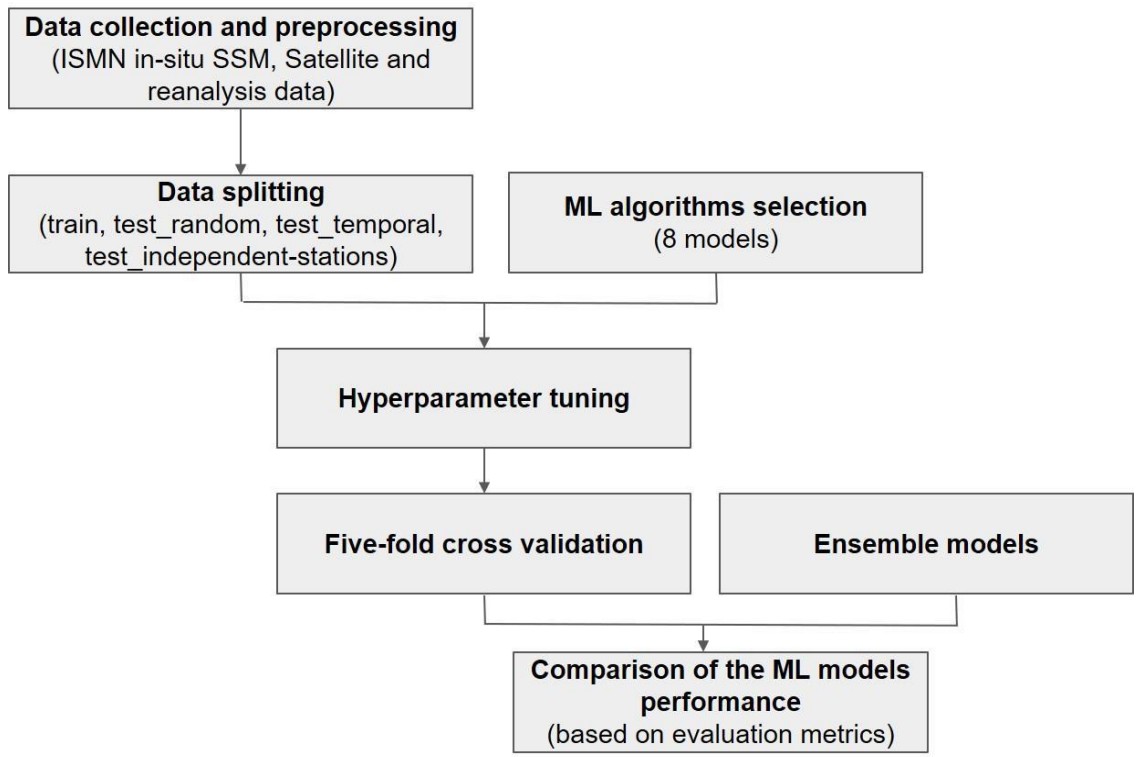

**Figure 2.** Conceptual framework of the optimisation operation of ML algorithms and construction of ensemble models.

The "Data collection and preprocessing" step was explained in Sections 2.1 and 2.2, while the "Data Splitting" operation was described in Section 2.3. The "ML algorithm selection" and "Hyperparameter Tuning" steps, related to the optimisation

procedure will be described in Section 3.1 and 3.2. The procedure to build the "Ensemble Models" is described in Section 3.3. The metrics considered for the comparison are presented in Section 3.4. Finally, the "Comparison of the ML models performance" (either optimised ML models, or ensemble models) is approached from different perspectives in Section 4 and 5.

### 3.1 Selection of Machine Learning Algorithms

We used the scikit-learn Python library (Pedregosa et al. 2011) to build and test the machine learning algorithms. Eight algorithms were selected based on their popularity and their proved performance in regression tasks (Sarker 2021). The algorithms considered for optimisation were (1) Random Forest Regressor (RFR) (Belgiu and Drăguţ 2016; Breiman 2001), (2) K-neighbours Regressor (KNR) (Papadopoulos et al. 2011), (3) AdaBoost (AB) (Yıldırım et al. 2019), (4) Stochastic Gradient Descent Regressor (SGDR), (5) Multiple Linear Regressor (MLR), (6) Multi-layer Perceptron Regressor (MLPR)

(Gaudart et al. 2004), (7) Extreme Gradient Boosting (XB) (Karthikeyan and Mishra 2021), and (8) GradientBoosting (GB) (Wei et al. 2019).



## 3.2 Optimisation Procedure (Hyper-Parameter tuning)

Each of the considered eight ML algorithms has different, specific parameters that can be tuned to improve the performance of the prediction. To optimise the training procedure and achieve the maximum performance of the algorithms for our task,
we applied the hyper-parameter tuning technique (Feurer and Hutter 2019), as it is one of the most popular methods to search for the best parameter values. In this study, grid search cross-validation (GridSearchCV) function (LaValle et al. 2004) was implemented in scikit-learn for hyper-parameter tuning. In the case of the ML algorithms that need long computation time, "itertools" Python function was applied (based on a for-loop).

The goal was to identify the best set of specific parameters of the eight considered ML algorithms, with the $r^2$ score used as
the evaluation metric. In the hyper-parameter tuning operation, we set a range for every parameter considered in various iterations. For example, for the 'n_neighbors' hyperparameter (specific to the KNR algorithm), we first set the possible values to 5, 10, 15; after we identified that the best set value is 5, we updated the range of possible values to 3, 4, 5, 6, 7 to narrow down the possible interval, and achieve the parameter values delivering more accurate performance metrics.

**Table 2.** Hyperparameters values explored in the tuning process for the optimisation of the considered machine learning
algorithms, the identified best combinations and their the $r^2$ score, and the training time of the optimisation procedure.

| ID | ML algorithm | Parameters considered in the optimisation procedure | Combination of parameters that achieved the best performance metrics | $r^2$ score value achieved by the best combination of parameters (on train set) | Training time of the tuning procedure (mins) |
|---|---|---|---|---|---|
| A1 | RFR | 'n_estimators': [start=10, stop=500, num=10] 'max_features' = ['sqrt', 'log2'] 'max_depth': [5, 6, 7, 9] 'min_samples_split': [2, 3, 4] 'min_samples_leaf': [1, 2, 4] | ['n_estimators': 10, 'max_features': 'log2' 'max_depth': None, 'min_samples_split': 4 'min_samples_leaf': 2] | 0.8590 | 30 |
| A2 | KNR | 'n_neighbors': [3,4,5,6,7] 'weights': ['uniform','distance'] 'p': [1,2] 'leaf_size': [20,30,40] 'algorithm': [ 'auto','ball_tree'] | ['n_neighbors': 4, 'weights': 'distance', 'p': 1, 'leaf_size': 20, 'algorithm': 'ball_tree'] | 0.8848 | 297 |



| A3 | AB | 'estimator': [DecisionTreeRegressor] 'estimator__max_depth': [3, 5, 10] 'estimator__criterion': ["squared_error", "friedman_mse", "poisson"] 'n_estimators': [10,20,30,40,50] 'learning_rate': [0.2,0.4,0.6,0.8,1.0] 'loss': ['linear'] | ['estimator':DecisionTreeRegressor, 'estimator__max_depth': 10, 'estimator__criterion':" squared_error", 'n_estimators':30, 'learning_rate':0.2, 'loss':'linear'] | 0.6547 | 50 |
|----|----|----|----|----|----|
| A4 | SGDR | 'loss': ['huber', 'epsilon_insensitive', 'squared_epsilon_insensitive'] 'penalty': ['l2', 'l1', 'elasticnet'] 'learning_rate': ['invscaling', 'constant', 'optimal', 'adaptive'] 'average': [False, True] 'warm_start': [False, True] 'alpha': [$10^{-4}$, $10^{-3}$, $10^{-2}$, $10^{-1}$, $10^{-5}$] | ['alpha': 0.01, 'average': True, 'learning_rate': 'constant', 'loss': 'epsilon_insensitive', 'penalty': 'l2', 'warm_start': False] | 0.4140 | 69 |
| A5 | MLR | - | - | 0.4148 | 0.0054 |
| A6 | MLPR | 'activation': ['identity', 'logistic', 'tanh', 'relu'] 'learning_rate': ['adaptive', 'invscaling', 'constant'] 'tol': array([$10^{-4}$, $10^{-5}$, $10^{-6}$, $10^{-7}$]) | ['activation': 'tanh' 'learning_rate': 'adaptive' 'tol': $10^{-7}$ 'alpha': $10^{-6}$ 'early_stopping': False | 0.7638 | 75 |





| | | | | | |
|---|---|---|---|---|---|
| | | 'alpha': array([$10^0$, $10^{-1}$, $10^{-2}$, $10^{-3}$, $10^{-4}$, $10^{-5}$]) 'early_stopping': [True, False] 'hidden_layer_sizes' (number of neurons in the first hidden layer, neurons number in the second hidden layer, and so on): 'layers': [1, 2, 3, 4], 'neurons': [1, 5, 10, 15, 19, 25, 30, 38, 57, 76] | 'max_iter': 1000 'hidden_layer_sizes': (30, 30)] | | |
| A7 | XB | 'n_estimators': [400, 500, 600, 700, 800], 'max_depth': [3, 4, 5, 6, 7, 8, 9, 10], 'min_child_weight': [1, 2, 3, 4, 5, 6], 'gamma': [0, 0.1, 0.2, 0.3, 0.4, 0.5, 0.6] 'subsample': [0.6, 0.7, 0.8, 0.9], 'colsample_bytree': [0.6, 0.7, 0.8, 0.9] 'reg_alpha': [0.05, 0.1, 1, 2, 3] 'reg_lambda': [0.05, 0.1, 1, 2, 3] 'learning_rate': [0.01, 0.05, 0.07, 0.1, 0.2] | ['n_estimators': 800 'max_depth': 10 'min_child_weight': 1 'gamma':0 'subsample': 0.8 'colsample_bytree': 0.9 'reg_alpha':0.05 'reg_lambda':0.1 'learning_rate':0.1] | 0.9139 | 1521 |
| A8 | GB | 'n_estimators': [80, 90, 100, 110, 120] 'max_depth': [1, 2, 3, 4, 5] 'learning_rate': [0.1, 0.3, 0.5] | ['n_estimators': 120, 'max_depth': 5, 'learning_rate': 0.5} | 0.7977 | 17 |



**Abbreviations:** ML=Machine Learning, RFR=Random Forest Regressor, KNR=K-neighbours Regressor, AB=AdaBoost, SGDR=Stochastic Gradient Descent Regressor, MLR=Multiple Linear Regressor, MLPR=Multi-layer Perceptron Regressor, XB=Extreme Gradient Boosting, GB=GradientBoosting.

**Notes: (ID=A1)**: "n_estimators" is the number of trees in the forest; "max_depth" is the maximum depth of the tree;
"min_samples_split" is the minimum number of samples required to split an internal node; while "min_samples_leaf" represents the minimum number of samples required to be at a leaf node. **(ID=A2)**: "n_neighbors" is the number of neighbours to use by default for k-neighbours queries; "weights" is weight function used in prediction; "p" is the power parameter for the Minkowski metric; "leaf_size" is the leaf size passed to BallTree or KDTree, and can affect the speed of the construction and query, as well as the memory required to store the tree; while the "algorithm" represents the algorithm
used to compute the nearest neighbours. **(ID=A3)**: "estimator" is the base estimator from which the boosted ensemble is built, if set to "None", then the base estimator is DecisionTreeRegressor; "estimator__max_depth" is the maximum depth of the tree, if set to "None", then nodes are expanded until all leaves are pure or until all leaves contain less than min_samples_split samples; "criterion" is the function to measure the quality of a split; "n_estimators" is the maximum number of estimators at which boosting is terminated; "learning_rate" is the weight applied to each regressor at each
boosting iteration; while "loss" represents the loss function to use when updating the weights after each boosting iteration. **(ID=A4)**: "loss" is the loss function to be used; "penalty" represents the regularisation term to be used; "average", the averaging method selected when set to "True", it computes the averaged SGD weights across all updates and stores the result in the "coef_attribute"; "warm_start" when set to "True", the algorithm reuses the solution of the previous call to fit as initialization, otherwise, it just erases the previous solution; while "alpha" is the constant that multiplies the regularisation
term. **(ID=A6)**: "activation" is the activation function for the hidden layer; "tol" represents the tolerance for the optimisation; "alpha" is the strength of the L2 regularisation term; while "early_stopping" indicates whether to use early stopping to terminate training when validation score is not improving; 'hidden_layer_sizes' is a array-like of shape (,), and the ith element represents the number of neurons in the ith hidden layer. **(ID=A7)**: "n_estimators" is the number of gradient boosted trees; "max_depth" is the maximum tree depth for base learners; "min_child_weight" represents the minimum sum of
instance weight (hessian) needed in a child; "gamma" codifies the minimum loss reduction required to make a further partition on a leaf node of the tree, "subsample" is the subsample ratio of the training instance; "colsample_bytree" is the subsample ratio of columns when constructing each tree; "reg_alpha" is the L1 regularisation term on weights (xgb's alpha); "reg_lambda" is the L2 regularisation term on weights (xgb's lambda); while "learning_rate" represents the boosting learning rate. **(ID=A8)**: "n_estimators" is the number of boosting stages to perform; "max_depth" is the maximum depth of
the individual regression estimators; while "learning_rate" codifies the learning rate that shrinks the contribution of each tree by "learning_rate".





### 3.3 Construction of the Ensemble Models

In order to enhance capabilities of individual models, we exploited ensemble techniques which helped to identify the proper combination of models. This way, we ensembled a set of base models (the optimised versions of machine learning models

mentioned in Section 3.1, obtained by applying the optimisation procedure described in Section 3.2) using model stacking techniques to combine the predictions of the models and built a new model.

The reasoning behind using an ensemble was that by stacking multiple base models representing different hypotheses, we can find a better hypothesis that might not be contained within the hypothesis space of the individual models from which the ensemble is built (Cira et al. 2020). Three main aspects causing this situation were identified (Dieterich 2000): (1) having

insufficient input data (statistical), (2) difficulties for the learning algorithm to converge to the global minimum (computational), and (3) representational, where the function cannot be represented by any of the hypotheses proposed and modelled by the algorithms during training.

A popular form of stacking involves computing the outputs of base models, performing a prediction for each model, and averaging their predictions inside the ensemble. In this technique, each $m$ sub-model contributes equally to a combined

prediction, $y$, as defined in Eq. 7. The specific steps are: (1) generate weak learners, each with its own initial values (by training them separately), and (2) combine these models in an ensembling environment, where their predictions are averaged for every instance of the set to compute.

$$\overline{y}(x) = \frac{1}{M} \sum_{m=1}^{M} y_m(x) \tag{7}$$

We built ensemble models by combining base models as diversely as possible (an example is illustrated in Figure 3). This way, the optimised versions of single ML algorithms become the weak learners, or base models, within ensemble models

that will be constructed using model bagging methodology. We applied ensemble learning procedures to test different combinations of the algorithms with the highest performance metrics and study their impact on the predicting performance.

The base models are the five ML models that achieved the highest performance metrics after the optimisation procedure described in Section 3.2. The constructed ensemble model variants will contain all the possible combinations of the five base models, taken three at a time ($C_n^k = \frac{n!}{k!(n-k)!}, where\ n = 5, k = 3$). In this way, ten ensemble models will be obtained and

the performance of every ensemble model built will be studied in Section 4.3.2.





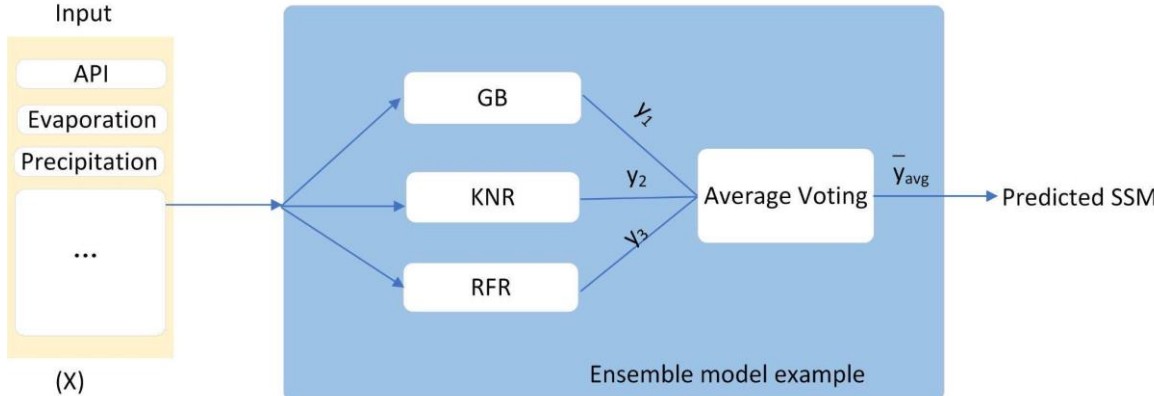

**Figure 3.** Example of an ensemble model structure (based on the average voting of three weak learners).

### 3.4. Evaluation of the Performance

We use five-fold cross validation to evaluate the training performance of different algorithms, both before and after the hyper-parameter tuning.

To assess the performances of different ML algorithms, we compared predicted SSM with in-situ observations. In this research, we considered three commonly used statistical evaluation metrics (Entekhabi et al. 2010b): the Root Mean Square Error (RMSE), defined in Eq. 8, the Pearson Correlation Coefficient (r) score, represented in Eq. 9, and the Coefficient of Determination R-square ($r^2$) score, presented in Eq. 10, as follows.

$$RMSE = \sqrt{\frac{\sum_{i=1}^{N}(y_{pred,i} - y_{ref,i})^2}{N}} \qquad (8)$$


$$r = \frac{\sum_{i=1}^{N}(y_{pred,i} - \overline{y_{pred,i}})(y_{ref,i} - \overline{y_{ref,i}})}{\sqrt{\sum_{i=1}^{N}(y_{pred,i} - \overline{y_{pred,i}})^2}\sqrt{\sum_{i=1}^{N}(y_{ref,i} - \overline{y_{ref,i}})^2}} \qquad (9)$$

$$r^2 = 1 - \frac{\sum_{i=1}^{N}(y_{ref,i} - y_{pred,i})^2}{\sum_{i=1}^{N}(y_{ref,i} - \overline{y_{ref,i}})^2} \qquad (10)$$

In Eqs. 8 to 10, $y_{pred,i}$ is the predicted SSM, $y_{ref,i}$ is the in-situ measured SSM, N is the number of valid pairs of SSM, $\overline{y_{pred,i}}$ is the mean value of the predicted SSM, $\overline{y_{ref,i}}$ is the mean value of the in-situ measured SSM.



In this study, there are three main steps, (1) the evaluation of the r² score on the training set, (2) the evaluation of the five-fold cross validation, and (3) the evaluation of the "test_random", "test_temporal", "test_independent-stations" sets. In the first step of the evaluation, the r² score on the training dataset was analysed to identify significant differences among different ML algorithms. In the second step, the r score, r² score, RMSE were computed to carry out the performance comparison on the "train" set using cross validation. In the third step, the r score, r² score, and RMSE were calculated on the

"test_random", "test_temporal", and "test_independent-stations" sets to compare the performance of the trained algorithms. The performance of the eight MLs and ten ensemble models were compared on the squared error of the predictions with non-parametric Kruskal-Wallis test at the 5% significance level, computed on the "test_random", "test_temporal" and "test_independent-stations" sets.

## 4. Results

### 4.1 Best Parameters from Hyper-parameter Tuning (Optimised Machine Learning Models)

The optimal performance after the hyperparameter tuning is presented in the fourth column of Table 2. The algorithms of RFR, KNR and XB show a superior result with r² scores of 0.859, 0.8848 and 0.9139 respectively, while the algorithms of GB, MLPR and ABGB also show a considerable result with r² scores of 0.7977, 0.7638, and 0.6547, respectively. It should be noticed that, in other relevant studies, such as (Kucuk et al. 2022), AB also performed slightly worse than RFR, GB and

XB in SSM estimation. However, the performance of the other two methods (SGDR and MLR) is not satisfactory. Both SGDR and MLR were unable to model the highly non-linear relationship between the soil moisture and the predictor variables because these two algorithms are linear regressors.

In the meanwhile, the computational efficiency of hyperparameter tuning of different algorithms varies greatly, sometimes with a different degree of magnitude. For example, it only took 17 minutes for GB to finish the turning, while XB needed

more than 25 hours. However, this seems to be more dependent on the choice of parameters and their range, as XB has nine parameters, and we selected at least five values to tune for each parameter.

### 4.2 Five-Fold Cross Validation

After hyper-parameters tuning, five-fold cross validation was used for performance comparison on the "train" dataset. The performance of the five-fold cross-validation for the eight algorithms is listed in Table 3. Similar to the result in Table 2, five

algorithms (RFR, KNR, MLPR, XB and GB) display a high performance. The XB achieved the best performance, with an RMSE of 0.0337 cm³/cm³ and an r² score of 0.9081. It is followed by the KNR algorithm, with an RMSE of 0.0392 cm³/cm³, and an r² score of 0.8760. RFR, GB and MLPR also achieved acceptable performance, even though the RMSE of 0.04 cm³/cm³ is often required in the community as the non-strict standard.

**Table 3.** Mean values and standard deviation of the evaluation metrics obtained in the five-fold cross validation.





| ID | ML algorithm | RMSE and standard deviation (cm$^3$/cm$^3$) | r score | r$^2$ score |
|----|--------------|--------------------------------------|---------|-------------|
| A1 | RFR  | 0.0413 ±0.0003  | 0.9228 ±0.0020 | 0.8627 ±0.0019 |
| A2 | KNR  | 0.0392 ±0.0003  | 0.9360 ±0.0011 | 0.8760 ±0.0021 |
| A3 | AB   | 0.0653 ±0.0003  | 0.8201 ±0.0017 | 0.6566 ±0.0025 |
| A4 | SGDR | 0.0853 ±0.0006  | 0.6435 ±0.0005 | 0.4140 ±0.0006 |
| A5 | MLR  | 0.0852 ±0.0006  | 0.6441 ±0.0005 | 0.4148 ±0.0006 |
| A6 | MLPR | 0.0541 ±0.00003 | 0.8741 ±0.0002 | 0.7638 ±0.0004 |
| A7 | XB   | 0.0337 ±0.0001  | 0.9533 ±0.0004 | 0.9081 ±0.0008 |
| A8 | GB   | 0.0500 ±0.0001  | 0.8935 ±0.0010 | 0.7982 ±0.0017 |

**4.3 Analysis of the Model Performance on "test_random", "test_temporal" and "test_independent-stations" Sets**

**4.3.1 Performance of Single, Optimised ML Models**

Having identified the best hyper-parameters for the considered algorithms and, therefore, the optimised versions of the ML models for our task, we next calculated and compared their performance on the "test_random", "test_temporal" and "test_independent-stations" sets. As described in Table 2, the amount of samples used for evaluating the performance (each containing nineteen predictor variables) for "train", "test_random", "test_temporal" sets were 352,075, 117,359, and 199,886 registries, respectively, together with 148 stations in the "test_independent-stations" set (containing 66,155 registries). The number of stations was 1574 for "train" & "test_random" and 1550 for "test_temporal" set.

From Table 4, we can find that RFR, KNR, XB performed better on the "test_random" set, compared to the other five. KNR achieved the best performance, achieving a maximum r$^2$ score of 0.8848. RFR, KNR, AB, XB, GB, MLPR performed relatively well on the "test_temporal" set, achieving a maximum r$^2$ score of 0.7126. In independent station evaluation, except MLPR, all ML algorithms perform similar but AB performs the best with r$^2$ score of 0.4905. In all, RFR, KNR, XB performed well in every step.

**Table 4.** Performance metrics obtained by eight optimised ML algorithms on the test sets.

| ID | ML algorithm | Test_random (117,359 registries) | | | Test_temporal (199,886 registries) | | | Test_independent-stations (66,155 registries) | | |
|----|--------------|-----------------------------|---------|-----------|-------------------------------|---------|-----------|-------------------------------------------|---------|-----------|
| | | RMSE (cm$^3$/cm$^3$) | r score | r$^2$ score | RMSE (cm$^3$/cm$^3$) | r score | r$^2$ score | RMSE (cm$^3$/cm$^3$) | r score | r$^2$ score |
| A | RFR | 0.0413 | 0.9301 | 0.8626 | 0.0599 | 0.8446 | 0.7126 | 0.0806 | 0.6869 | 0.4649 |





| 1 | | | | | | | | | |
|---|---|---|---|---|---|---|---|---|---|
| A 2 | KNR | 0.0379 | 0.9407 | 0.8848 | 0.0667 | 0.8099 | 0.6435 | 0.0900 | 0.6337 | 0.3327 |
| A 3 | AB | 0.0651 | 0.8222 | 0.6597 | 0.0696 | 0.7884 | 0.6114 | 0.0786 | 0.7005 | 0.4905 |
| A 4 | SGDR | 0.0853 | 0.6444 | 0.4152 | 0.0852 | 0.6483 | 0.4173 | 0.0854 | 0.6347 | 0.3995 |
| A 5 | MLR | 0.0852 | 0.6448 | 0.4158 | 0.0851 | 0.6491 | 0.4191 | 0.0852 | 0.6357 | 0.4018 |
| A 6 | MLPR | 0.0546 | 0.8723 | 0.7605 | 0.0640 | 0.8234 | 0.6710 | 0.1014 | 0.5724 | 0.1528 |
| A 7 | XB | 0.0385 | 0.9385 | 0.8806 | 0.0609 | 0.8414 | 0.7023 | 0.0817 | 0.6860 | 0.4499 |
| A 8 | GB | 0.0502 | 0.8932 | 0.7977 | 0.0617 | 0.8353 | 0.6948 | 0.0842 | 0.6657 | 0.4158 |

### 4.3.2 Performance of Ensemble Models

Based on the results displayed in Table 4, the ensemble regressors were built using RFR, KNR, XB, GB and AB as base models (as they displayed the best performance). We combined three ML algorithms in each combination to form the ten ensemble models. The performance of these ten models was found to be stable (as observed in Table 5). Specifically, KNR_RFR_XB and GB_RFR_XB displayed the best performance in "test_random" (RMSE were 0.0355 cm$^3$/cm$^3$ and 0.0391 cm$^3$/cm$^3$, and the r scores were 0.9488 and 0.9379) and test_temporal (RMSE were 0.0576 cm$^3$/cm$^3$ and 0.0568 cm$^3$/cm$^3$ and r scores were 0.8571 and 0.8614) sets. However, there were no considerable differences among the different combinations for test_independent-stations.

The voting regressors built with ensembling techniques generally showed improved performance when compared to the considered base models. For example, AB achieved a RMSE of 0.0651 cm$^3$/cm$^3$, 0.0696 cm$^3$/cm$^3$, and 0.0786 cm$^3$/cm$^3$, in





the "test_random", "test_temporal", and "test_independent-stations", respectively, while in the voting regressor result, the six combinations that had AB were able to achieve RMSEs of 0.0417 cm³/cm³ to 0.0468 cm³/cm³ in the "test_random" set, 0.0584 cm³/cm³ to 0.0593 cm³/cm³ in the "test_temporal" set and 0.0767 cm³/cm³ to 0.0775 cm³/cm³ in "test_independent-stations". The ensemble models improved AB's performance by 0.0183-0.0234 cm³/cm³ (28% - 36%), 0.0103-0.0112 cm³/cm³ (15-16%), and 0.0011-0.0019 cm³/cm³ (1.4%-2.4%) in the "test_random", "test_temporal" and "test_independent-stations" sets in terms of RMSE.

The best performing voting regressor (KNR_RFR_XB, composed of XB, RFR, and KNR) achieved a RMSE value of 0.0355 cm³/cm³ on the "test_random" set. The RMSE of KNR, RFR, and XB were 0.0379, 0.0413 and 0.0385 cm³/cm³ in "test_random", and the r scores of KNR, RFR, and XB were 0.9407, 0.9301, and 0.9385. Compared with KNR, RFR, and XB, the ensemble model improved, for RFR, 0.0058 cm³/cm³ (14%) of RMSE and 0.0187 (2%) of r score. On the "test_temporal" and "test-independent-stations", the performance of KNR_RFR_XB also performed better than the single three ML algorithms. In summary, models built with ensembling techniques averaged the performance of base ML algorithms and performed more stable than single ML algorithms. Overall, ensembling techniques improved the performance of base algorithms in "test_random" dataset, but had little effect on the "test_independent-stations" dataset.

**Table 5.** Performance metrics obtained by the ten ensemble models built with different combinations of selected machine learning algorithms.

| ID | Combination of MLs in the ensemble models | Test_random (117,359 registries) | | | Test_temporal (199,886 registries) | | | Test_independent-stations (66,155 registries) | | |
|---|---|---|---|---|---|---|---|---|---|---|
| | | RMSE (cm³/cm³) | r score | r² score | RMSE (cm³/cm³) | r score | r² score | RMSE (cm³/cm³) | r score | r² score |
| E1 | GB_KNR_RFR | 0.0389 | 0.9386 | 0.8783 | 0.0575 | 0.8580 | 0.7347 | 0.0778 | 0.7099 | 0.5019 |
| E2 | AB_RFR_XB | 0.0429 | 0.9298 | 0.8518 | 0.0587 | 0.8531 | 0.7234 | 0.0771 | 0.7153 | 0.5108 |
| E3 | GB_KNR_XB | 0.0383 | 0.9402 | 0.8821 | 0.0572 | 0.8599 | 0.7374 | 0.0781 | 0.7088 | 0.4980 |
| E4 | GB_RFR_XB | 0.0391 | 0.9379 | 0.8772 | 0.0568 | 0.8614 | 0.7410 | 0.0767 | 0.7198 | 0.5154 |
| E5 | KNR_RFR_XB | 0.0355 | 0.9488 | 0.8985 | 0.0576 | 0.8571 | 0.7335 | 0.0775 | 0.7122 | 0.5046 |
| E6 | AB_GB_KNR | 0.0457 | 0.9189 | 0.8323 | 0.0588 | 0.8526 | 0.7225 | 0.0775 | 0.7114 | 0.5055 |
| E7 | AB_GB_RFR | 0.0468 | 0.9144 | 0.8237 | 0.0591 | 0.8516 | 0.7196 | 0.0774 | 0.7124 | 0.5055 |





| E8 | AB_GB_XB | 0.0462 | 0.9161 | 0.8282 | 0.0585 | 0.8541 | 0.7255 | 0.0767 | 0.7184 | 0.5153 |
| E9 | AB_KNR_RFR | 0.0426 | 0.9310 | 0.8542 | 0.0593 | 0.8497 | 0.7177 | 0.0768 | 0.7168 | 0.5137 |
| E10 | AB_KNR_XB | 0.0417 | 0.9342 | 0.8605 | 0.0584 | 0.8544 | 0.7266 | 0.0774 | 0.7121 | 0.5066 |

**Abbreviations:** RFR=Random Forest Regressor, KNR=K-neighbours Regressor, AB=AdaBoost, SGDR=Stochastic Gradient Descent Regressor, MLR=Multiple Linear Regressor, MLPR=Multi-layer Perceptron Regressor, XB=Extreme Gradient Boosting, GB=GradientBoosting. **Notes:** (1) GB_KNR_RFR refers to the ensemble model constructed by GB, KNR and RFR; while AB_RFR_XB refers to the ensemble model constructed by AB, RFR and XB. (2) The same naming procedure applies for GB_KNR_XB, GB_RFR_XB, KNR_RFR_XB, AB_GB_KNR, AB_GB_RFR, AB_GB_XB, AB_KNR_RFR, AB_KNR_XB.

### 4.3.3 Comparison of Single, Optimised ML and Ensemble Models

Table 6 shows which methods performed significantly better or worse and the cumulative rank of the methods based on the Kruskal-Wallis order computed for the three test sets. Based on the analysis on the "test_random" set, KNR is significantly more accurate than the other methods. KNR ensembled with RFR and XB, or RFR and GB, or GB and XB are the second, third and fourth most accurate methods, respectively. When the performances of single algorithms are considered, RFR, XB, and GB follow the performance of KNR. MLR and SGDR display the weakest performance. If AB which has a weaker performance as a single ML is ensembled with any of the following algorithms: KNR or XB or RFR – which are all significantly more accurate than AB, the prediction performance of the ensemble significantly decreases when compared to the performance of those single algorithms. GB, MLPR, AB, SGDR and MLR describe the relationship between the target and predictors significantly less accurately than the ensemble methods.

On "test_temporal" set, KNR_RFR_XB performs significantly better than any of the other methods. GB ensembled with KNR and XB or KNR and RFR or RFR and XB are the second, third and fourth most accurate methods. Among the single ML methods RFR is significantly the most accurate, which is followed by KNR, XB, GB, MLPR. Similarly to the results on "test_random" set, MLR and SGDR performed significantly worse when compared to any other method. If AB is ensembled with KNR or RFR, the prediction performance does not improve significantly or significantly decreases compared to the performance of those single MLs.

On "test_independent-stations" set, the best performing predictions could be reached by using the ensemble of any three MLs from KNR, RFR, XB, and GB. Among the single, optimised ML methods, RFR is significantly the most accurate, and is followed by XB, AB, GB, KNR, MLR, SGDR, MLPR, respectively. All ensembles perform significantly better than the single MLs.

The results underpin that the inclusion of a not well performing algorithm in an ensemble will lead to a worse performance than a single well performing model.





**Table 6.** Result of the Kruskal-Wallis test.

| ID | Method | Significant difference | | | Rank based on sign. diff. analysis |
|----|--------|------------------------|--|--|------------------------------------|
| | | Test_random (117,359registries) | Test_temporal (199,886 registries) | Test_independent-stations (66,155 registries) | |
| E5 | KNR_RFR_XB | l | m | i | 4 |
| E3 | GB_KNR_XB | j | l | hi | 8 |
| E1 | GB_KNR_RFR | k | k | hi | 8 |
| E4 | GB_RFR_XB | i | k | i | 9 |
| A1 | RFR | j | j | f | 13 |
| E10 | AB_KNR_XB | h | i | gh | 14 |
| A2 | KNR | m | i | c | 15 |
| E9 | AB_KNR_RFR | h | fg | gh | 17 |
| A7 | XB | i | h | ef | 17 |
| E6 | AB_GB_KNR | f | g | gh | 18 |
| E2 | AB_RFR_XB | g | g | g | 18 |
| E8 | AB_GB_XB | e | f | hi | 20 |
| E7 | AB_GB_RFR | e | e | g | 23 |
| A8 | GB | d | d | d | 29 |
| A3 | AB | b | b | e | 32 |
| A6 | MLPR | c | c | a | 34 |
| A5 | MLR | a | a | b | 37 |
| A4 | SGDR | a | a | b | 37 |



Notes: Different letters show that there is a significant difference between the methods. Letter "a" indicates the worst
performing algorithm.

Based on the result of the Kruskal-Wallis test, the best single ML and best ensemble algorithms were identified. The
probability density function (PDF) of them are shown in Figure 4. On "test_random" set, the overlap between in-situ SSM
and predicted SSM from KNR and KNR_RFR_XB are 96.2% and 93.1%. On "test_temporal" set, the overlap between in-

situ SSM and predicted SSM from RFR and KNR_RFR_XB are 87.9% and 90.7%. On "test_independent-stations" set, the
overlap between in-situ SSM and predicted SSM from RFR and KNR_RFR_XB are 80.5% and 81.0%. It shows that the
ensemble model performs better on unseen data.

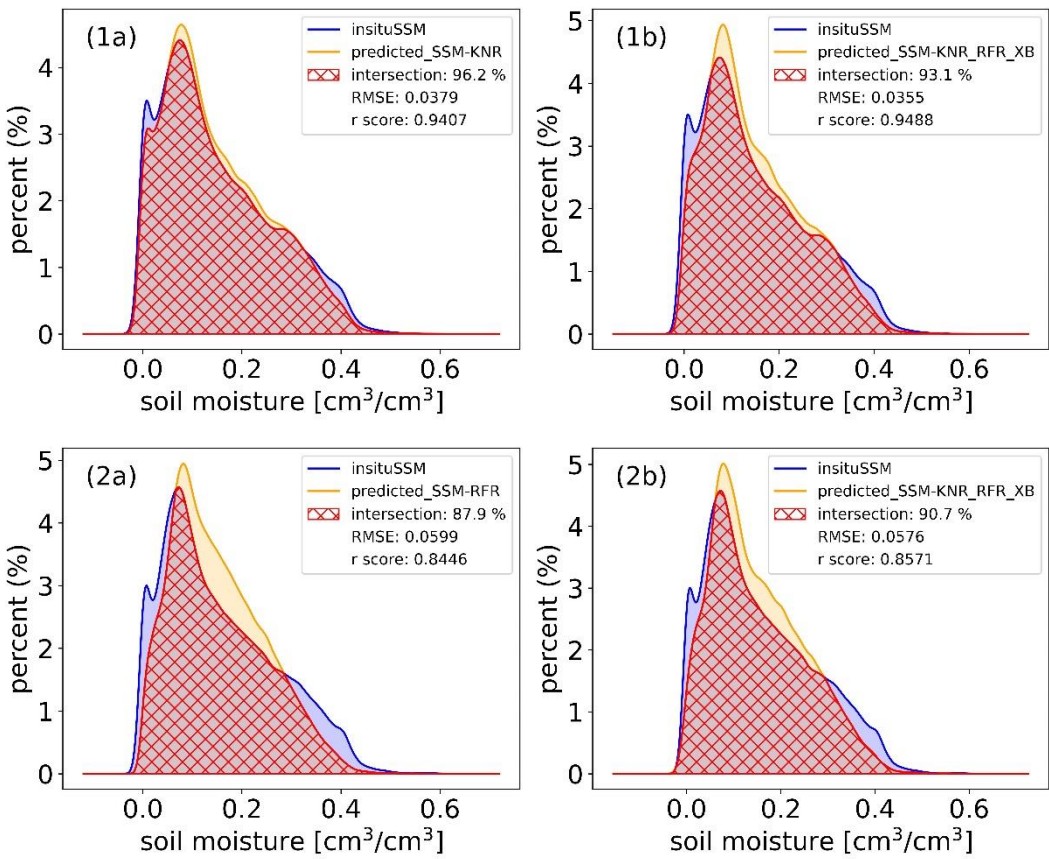



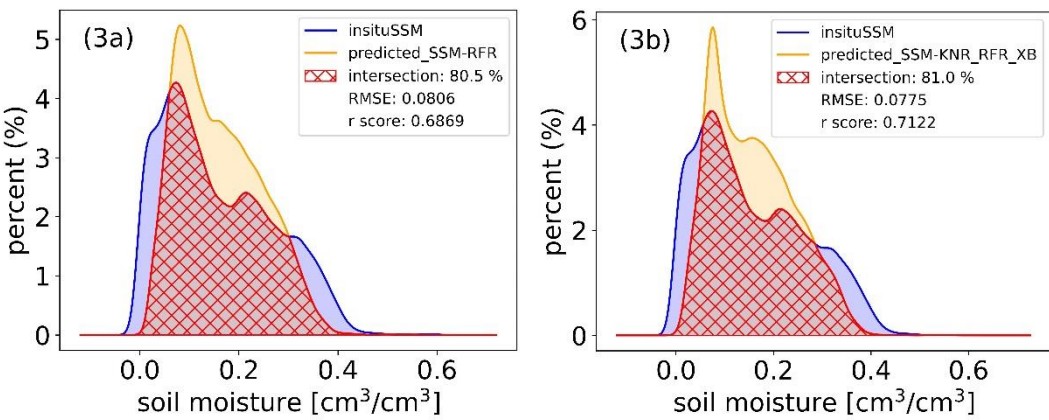


**Figure 4.** In-situ SSM with predicted SSM from the best single, optimised ML model (**1a**) and best ensemble model (**1b**) on the "test_random" set (KNR, KNR_RFR_XB); in-situ SSM with predicted SSM from the best single, optimised ML (**2a**) and best ensemble algorithm (**2b**) on the "test_temporal" set (RFR, KNR_RFR_XB); in-situ SSM with predicted SSM from the best single, optimised ML (**3a**) and best ensemble algorithm (**3b**) on the "test_independent-stations" set (RFR,
KNR_RFR_XB).

### 4.4 Performance on the "test_independent-stations" Set Grouped by Climate Zones

In Section 4.3, we analysed the performance of our model on three sets, "test_random", "test_temporal", and "test_independent-stations", which consist of stations located in different climate zones and, as expected, we observed important variations in the model's performance across different stations (due to their unique climatic condition).

To further examine  the performance of each station in the "test_independent-stations" set, we considered the number of stations in both the "train" and "test_independent-stations" sets. Specifically, the stations from the "train" set are distributed across nineteen Köppen climate zones, while the "test_independent-stations" set stations are across sixteen climate zones (Cwb, Dsc, Dwb, are not represented in  the "test_independent-stations" set).

To ensure a robust climate zones analysis, we excluded stations with less than 100 days of observations in the
"test_independent-stations" set. Figure 5 shows the number of stations on the  "test_independent-stations" dataset over each climate zone (the number of stations decreased from 148 to 117 after this applying this (100 days) filter and the climate zones covered by "test_independent-stations" set decreased from sixteen to fifteen, Dwc being removed).





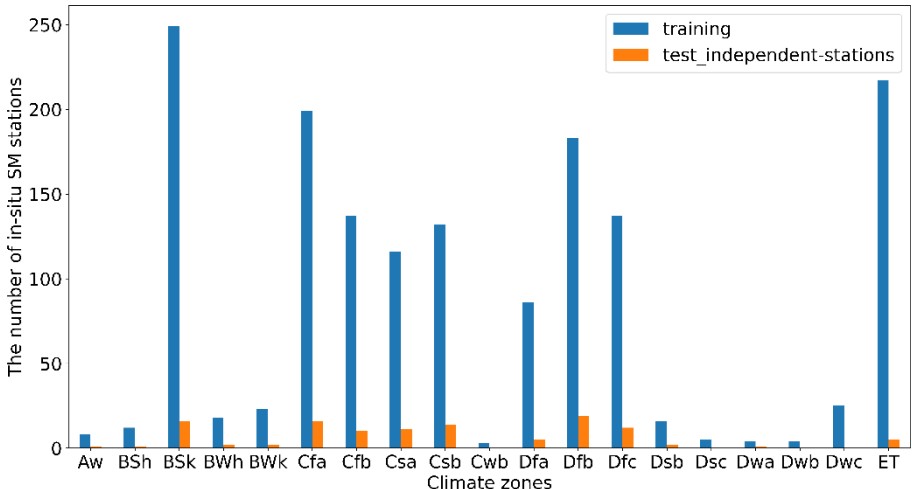

**Figure 5.** Number of in-situ SSM stations in "train" set and "test_independent-stations" set over each climate zone.

**Abbreviations:** Aw (Tropical wet and dry or savanna climate), BSh (Hot semi-arid climate), BSk (Cold semi-arid climate), BWh (Hot desert climate), BWk (Cold desert climate), Cfa (Humid subtropical climate), Cfb (Temperate oceanic climate), Csa (Hot-summer Mediterranean climate), Csb (Warm-summer Mediterranean climate), Cwb (Subtropical highland climate), Dfa (Hot-summer humid continental climate), Dfb (Warm-summer humid continental climate), Dfc (Subarctic climate), Dsb (Mediterranean-influenced warm-summer humid continental climate), Dsc (Mediterranean-influenced subarctic climate),

Dwa (Monsoon-influenced hot-summer humid continental climate), Dwb (Monsoon-influenced warm-summer humid continental climate), Dwc (Monsoon-influenced subarctic climate), and ET (Tundra climate).

The performances of the eight optimised ML algorithms and ten constructed ensemble models in the fifteen climate zones are described in the boxplots of Figure 6 (which includes the stations from the "test-independent-stations" set with more than

100 days of in-situ measurements). In Figure 6a, the median RMSE values in most of the climate zones were below 0.1 cm$^3$/cm$^3$.. In Figure 6b, except BSh, BWh, in other thirteen climate zones, we can find at least one single ML algorithm that can have a median r score higher than 0.6. GB, MLPR, SGDR and RFR can achieve a median r score of 0.6 in twelve, eleven, nine, and nine climate zones, respectively. The median of r score from GB in BSh, BWh and BWk were below 0.6 because these three climate zones all have sparse "train" stations and they are all arid climates. In Figure 6c, the median

values of RMSE in all climate zones were below 0.075 cm$^3$/cm$^3$ in the case of ensemble models. Figure 6d, we can see that the r scores in other thirteen climate zones were above 0.6 except BSh and BWh. This indicates that the number of "train" stations also plays an important role in ensemble models. Ensemble models can improve the performance but can not completely solve the problem of lacking training data.

Using ensemble models is a way to improve the performance of the single, optimised ML algorithms. However, the outliers

in the lower part of the r score boxplot were still present in BSk and Dfb, even for the ensemble models.



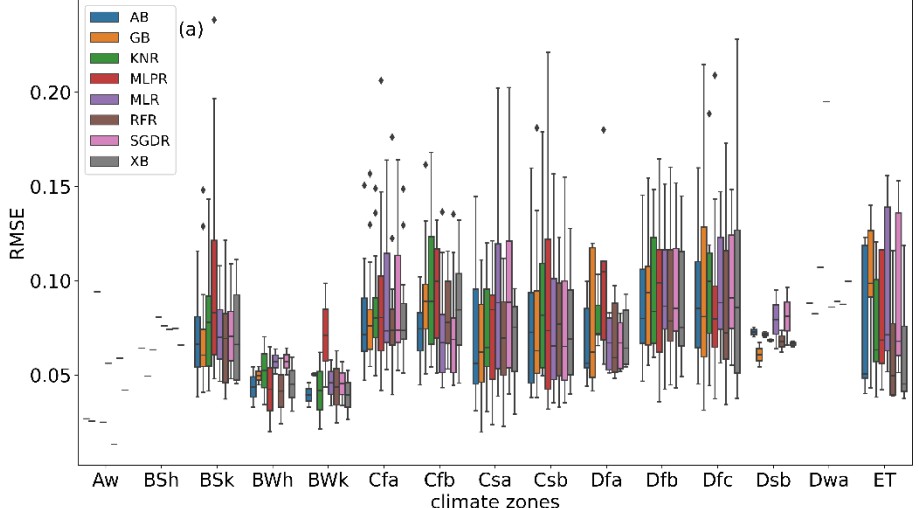

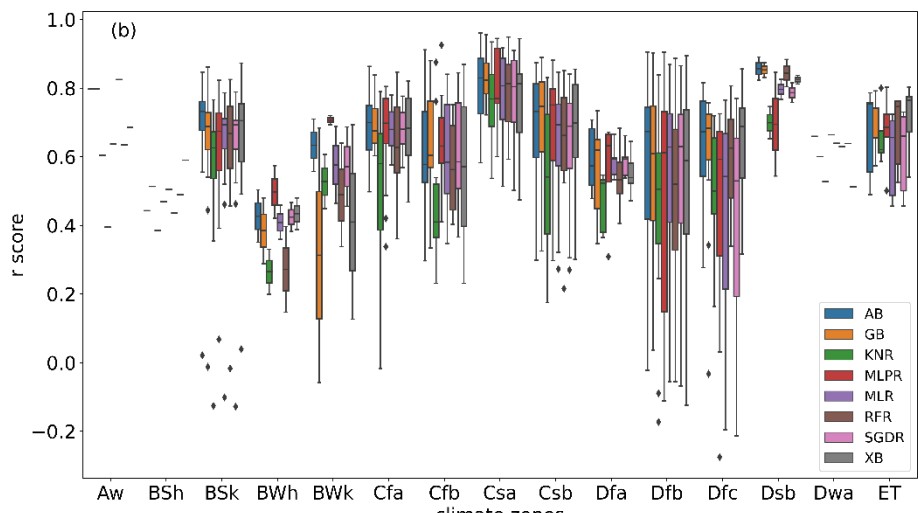



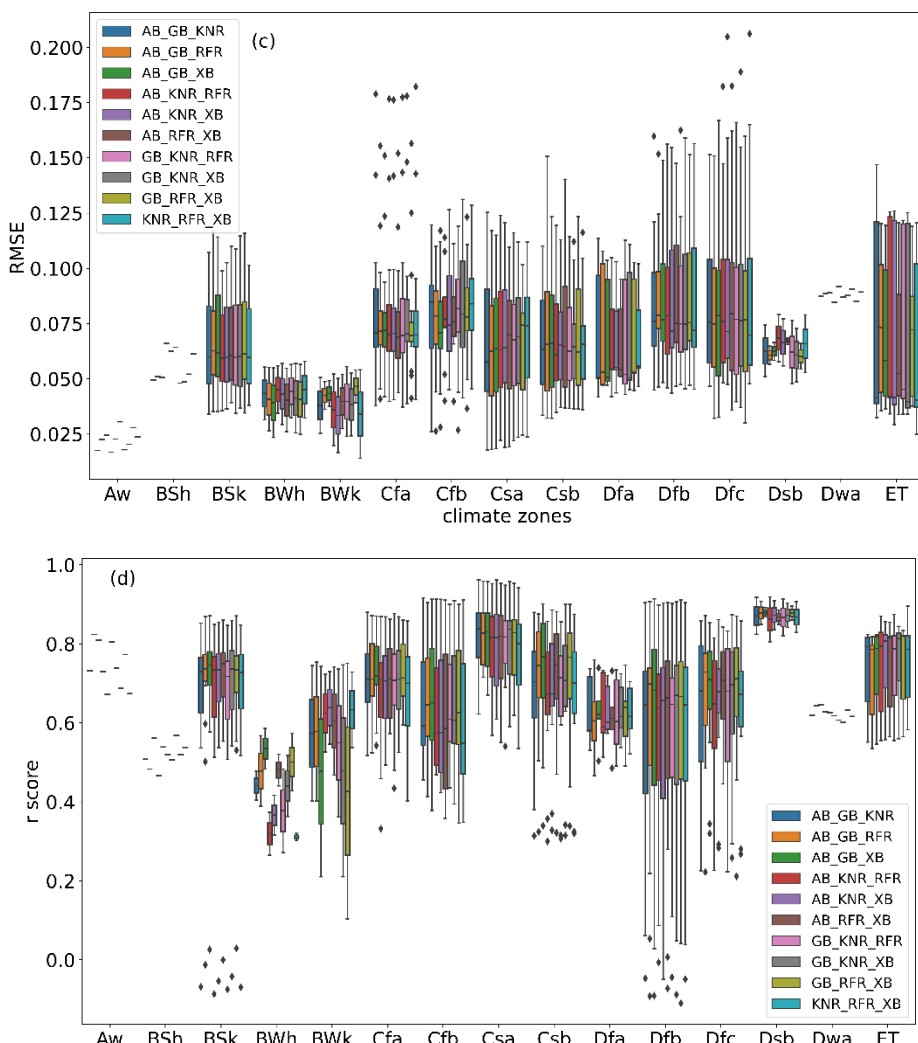

**Figure 6.** Boxplot of RMSE and r score for "test_independent-stations" set based on climate zones -the performance of eight optimised ML algorithms can be found in **(a)** and **(b)**, while the performance of ten ensemble models are shown in **(c)** and **(d)**.

**Abbreviations: (1)** Related to the climate zones: Aw (Tropical wet and dry or savanna climate), BSh (Hot semi-arid climate), BSk (Cold semi-arid climate), BWh (Hot desert climate), BWk (Cold desert climate), Cfa (Humid subtropical climate), Cfb (Temperate oceanic climate), Csa (Hot-summer Mediterranean climate), Csb (Warm-summer Mediterranean climate), Dfa (Hot-summer humid continental climate), Dfb (Warm-summer humid continental climate), Dfc (Subarctic climate), Dsb (Mediterranean-influenced warm-summer humid continental climate), Dwa (Monsoon-influenced hot-summer humid continental climate), ET (Tundra climate). **(2)** Related to the ML algorithms: RFR=Random Forest Regressor ,





KNR=K-neighbours Regressor, AB=AdaBoost, SGDR=Stochastic Gradient Descent Regressor, MLR=Multiple Linear
Regressor, MLPR=Multi-layer Perceptron Regressor, XB=Extreme Gradient Boosting, GB=GradientBoosting.

**Notes: (1)** GB_KNR_RFR refers to the ensemble model constructed by GB, KNR and RFR; while AB_RFR_XB refers to the ensemble model constructed by AB, RFR and XB. **(2)** The same naming procedure applies for GB_KNR_XB, GB_RFR_XB, KNR_RFR_XB, AB_GB_KNR, AB_GB_RFR, AB_GB_XB, AB_KNR_RFR, AB_KNR_XB.

### 4.5 Performance on Single, Selected Stations from the "test_independent-stations" Set

We selected the BSk climate zone as an example to deepen the analysis of outliers. The weakest r scores were present at station "SEVILLETA" (lon: 106.68° W, lat: 34.36° N), the r scores being all lower than 0.4 for each of the eight single, optimised ML algorithms and the ten ensemble models.

For our single-station analysis, we also chose two stations where the performance values were improved by the ensemble models. In Figure 7, we can observe that all r scores computed for the  ensemble models at station CARRIZORAN (lon:
120.03° W, lat: 35.28° N, in climate zone BSk) were above 0.8, while at station WHISKEYCK (lon: 105.12° W, lat: 37.21° N, in climate zone BSk) all r scores were above 0.6.

In Figure 8, it can be found that the predicted SSM was consistent with the in-situ SSM at both of these two stations. For the outlier station "SEVILLETA", we found that the r score between API and predicted SSM in station SEVILLETA is 0.8, and the r score between evaporation and predicted SSM is -0.67. This implied that our predictor variables were consistent with
our predicted SSM, but the   predictor variables were not the main decisive factor for the in-situ SSM in station "SEVILLETA". There was no significant difference between the main static predictor variables of station "SEVILLETA" and other stations in the BSk climate zone. The potential cause could be an environmental factor that is not considered as an independent variable during the prediction (e.g. different vegetation types).





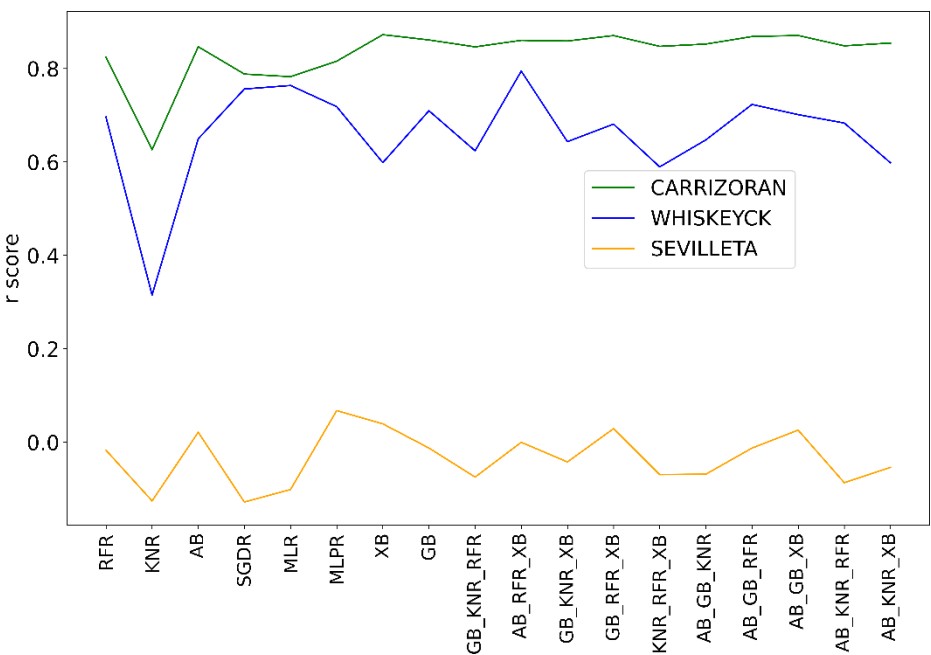

**Figure 7.** The r scores of three stations obtained by eight optimised ML algorithms and ten ensemble models.

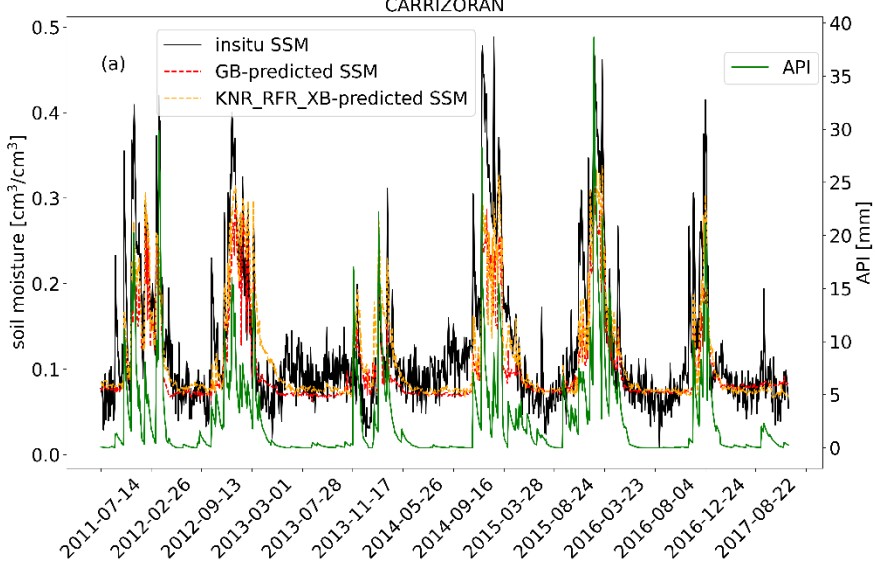





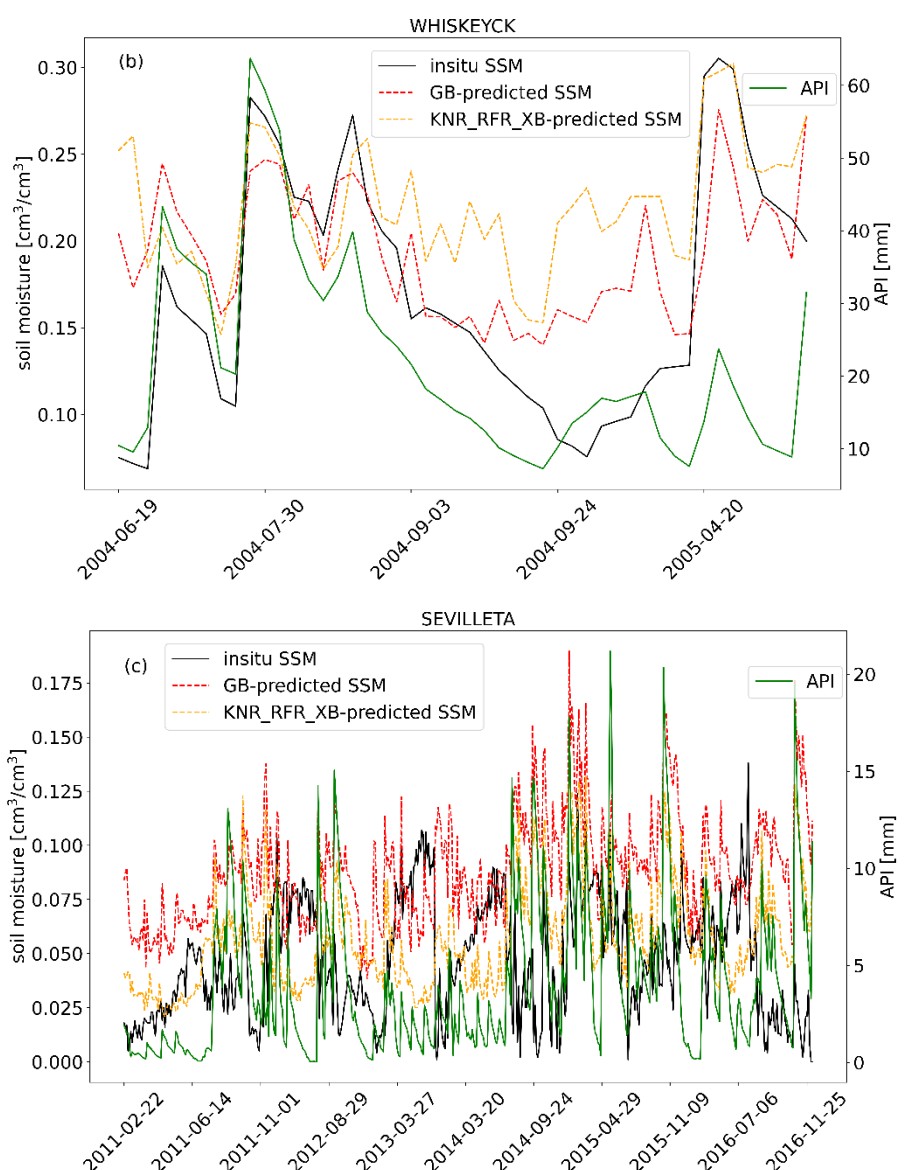



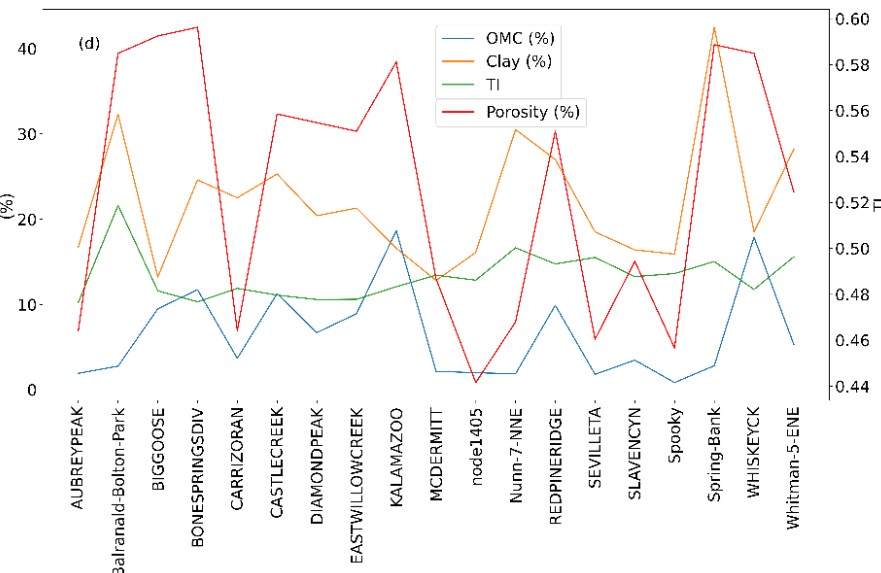

**Figure 8.** In-situ and predicted SSM from GB and KNR_RFR_XB and API in station **(a)** CARRIZORAN, **(b)** WHISKEYCK and **(c)** SEVILLETA; **(d)** static predictors of stations in climate zone BSk.

## 5. Discussion

Soil moisture is an important parameter for understanding the water cycle, predicting crop yields, and managing irrigation. Remotely sensed soil moisture has become an important dataset for supporting many human activities, such as water
resource conservation and management, environmental monitoring, and disaster response (Su et al. 2003). In this section, we will discuss the performance and the uncertainty of single, optimised ML algorithms and the proposed ensemble models.

### 5.1 Performance of Trained Regression Models

This study presents one of the first studies constructing and comprehensively evaluating different ML algorithms for estimating SSM with not only optimised, individual models, but also with ensemble models using the optimised algorithms
as base models. The predictor variables were selected based on their relevance to physical processes in the land-atmosphere interaction.

First, based on various predictor variables extracted from multi-source datasets, we trained and tested eighteen ML models in order to find their optimised version for estimating SSM. The cross-validation result showed that the RFR, KNR and XB outperform other ML algorithms. When evaluated on "test_random" set, KNR achieved the lowest RMSE of 0.0379 cm$^3$/cm$^3$
among individual ML algorithms and a high r score of 0.9407. While on "test_temporal", RFR showed the lowest RMSE of 0.0599 cm$^3$/cm$^3$ and highest r score of 0.8446. On "test_independent-stations", AB showed the lowest RMSE 0.0786 cm$^3$/cm$^3$ and highest r score of 0.7005.




Second, the ensemble models improved the performance of the individual ML algorithms if the ensemble does not include a significantly worse performing algorithm. From Section 4.3.2, we can observe that ensemble models mostly improved the performance of base algorithms in "test_random" sets, and had the minimal impact on "test_independent-stations". From section 4.3.3, the ensemble models based on the combination of KNR_RFR_XB showed the best performance on "test_random" and "test_temporal". KNR_RFR_XB  and GB_RFR_XB had the significantly best performance on "test_independent-stations" set. Considering the performance of all eight MLs and ten ensembles on all three test sets the ensemble of KNR and RFR and XB performs the best, thus this is the suggested method to predict SSM based on the presented predictors.

Third, we found significant variations in the model's performance across different stations because of their unique climatic conditions. Therefore, we further analysed the "test_independent-stations" performance on climate zone level. For single, optimised ML algorithms, the median RMSE from five models except MLPR, MLR and SGDR were below 0.1 cm$^3$/cm$^3$. GB, XB and RFR achieved a median r score of 0.6 in twelve, eleven, and nine climate zones, respectively, out of fifteen climate zones. The ensemble models improved significantly the performance. The median value of RMSE in all climate zones were all below 0.10 cm$^3$/cm$^3$, and all voting regressors achieved the r scores above 0.6 in thirteen climate zones except BSh and BWh because of the sparse distribution of "train" stations.

Forth, we selected the BSk climate zone as an example to deepen the analysis of outlier stations. The result showed that ensemble models can improve the performance of single ML algorithms and obtain more stable results. The r scores from ensemble models at station CARRIZORAN were all above 0.8, and those at station WHISKEYCK were all above 0.6, while, in general, the optimised individual ML algorithms did not perform as good as the ensemble models. However, we found the worst performance at the outlier station SEVILLETA that the ensemble models could not improve. The potential cause could be an environmental factor that is not considered as an independent variable during the prediction (e.g. different vegetation types). Another potential cause could be the non-representativeness of the in-situ measurements for the studied pixel scale.

- DISCUSS ADVANTAGES OF THE ENSEMBLE:

Advantage 1:  The ensemble models significantly improve the accuracy of the individual ML algorithms. By combining multiple regression models, the ensemble models can capture more of the complexity and variability in the data, leading to more accurate predictions (Cira et al. 2020).

Advantage 2: Because the ensemble models uses multiple models, they are less sensitive to outliers or individual model errors, making them more robust and stable models.

Advantage 3: The ensemble models allow us to use different types of regression models and hyperparameters, so researchers can choose the best models for our particular data and problem.

Advantage 4: The ensemble models can reduce overfitting by combining models that have different strengths and weaknesses, leading to a more generalizable model.



- DISCUSS DISADVANTAGES ENSEMBLE:

Disadvantage 1: The training of algorithmic implementations within ensembling environments requires more computational power. However, the increased and the more stable prediction behaviour is more desirable when tackling tasks where high performance metrics are expected.

Disadvantage 2: Depending on the base models, the performance of ensemble models can sometimes worsen when compared to well-performing optimised algorithms. For this reason, it is advised to optimise the base algorithms as much as possible for the chosen task. It is also observed that regression algorithms with a higher complexity generally displayed a higher generalisation capacity.

## 5.2 Uncertainty of Proposed Models

There is often uncertainty in these remotely sensed soil moisture dataset due to several factors. One of them comes from the variability of the soil itself. For instance, soil properties such as texture, and organic matter content can affect the ability of the soil to hold water and how quickly it absorbs or releases moisture. Such spatial variability of soil properties can lead to differences in the observed soil moisture even within a small area. Lots of existing soil moisture products in large areas are obtained from point measurements over heterogeneous landscape which leads to uncertainty in the estimation at the pixel scale (Guerschman et al. 2015). To reduce this uncertainty, we used relatively high spatial resolution that provides more detailed and accurate information about the distribution of soil moisture across different climate regions at a global scale. We comprehensively explored different ML models working on multi-source datasets, including ERA5-Land, MODIS, Soilgrids, and MERIT Hydro.

Specifically, we used high-resolution remotely-sensed products and reanalysis data at the global scale, so that it is possible to generate detailed maps of soil moisture at a fine spatial resolution, which can help to reduce uncertainty due to spatial heterogeneity and improve the accuracy of soil moisture estimates.

We also revealed that special care should be taken for areas with high uncertainty when applying the ML models. For instance, the areas with less ground samples may have less representation during the training process, e.g. training data includes low number of samples from climate zones BSh, BWh and BWk, thus application of the presented method might result in higher uncertainty than in climate zones where training data is more representative in terms of land surface and soil properties.

Some other aspects should also receive attention. For instance, hyper-parameters tuning is a computationally expensive operation that proved to have an important effect on the performance of each machine learning model. However, it involves the human factor with expertise in choosing the right ranges for each hyperparameter in order to achieve the best possible training. We recommend to carry out the training in at least two iterations, first selecting wider parameter intervals, and then narrowing it down to ranges in proximity to the best value detected in the initial experiments.



## 6. Conclusions

Soil moisture plays an essential role in the exchanges of water, energy and carbon between land and the atmosphere. In this study, we investigated performance of different ML algorithms in estimating SSM based on data of the International Soil Moisture Network (ISMN) collected from 1722 stations with 8 Machine Learning algorithms and 10 ensemble models. The major findings of this study are as follows.

The algorithms considered for optimisation were (1) Random Forest Regressor (RFR) , (2) K-neighbours Regressor
(KNR)(Papadopoulos et al. 2011), (3) AdaBoost (AB) (Yıldırım et al. 2019), (4) Stochastic Gradient Descent Regressor (SGDR), (5) Multiple Linear Regressor (MLR), (6) Multi-layer Perceptron Regressor (MLPR) (Gaudart et al. 2004), (7) Extreme Gradient Boosting (XB) (Karthikeyan and Mishra 2021), and (8) GradientBoosting (GB)(Wei et al. 2019).

The cross-validation result showed that the Random Forest Regressor (RFR), K-neighbours Regressor (KNR) and Extreme Gradient Boosting (XB) outperform other ML algorithms. From Kruskal-Wallis test result, KNR performs best on
"test_random" set, while RFR performs best on "test_temporal" and "test_independent-stations". The ensemble models proved to improve the performance of the individual ML algorithms, the best performing ensemble model in "test_random", "test_temporal" and "test_independent-stations" is the KNR_RFR_XB.

The optimised ML algorithms achieved the median RMSEs of below 0.1 $cm^3/cm^3$ on the "test_independent-stations" on climate zone level. GradientBoosting (GB), Multi-layer Perceptron Regressor (MLPR), Stochastic Gradient Descent
Regressor (SGDR), and Random Forest Regressor (RFR) achieved a median r score of 0.6 in twelve, eleven, nine and nine climate zones, respectively, out of fifteen climate zones. The ensemble models improved the performance significantly, with the median value of RMSE in all climate zones all below 0.075 $cm^3/cm^3$. All voting regressors achieved the r scores of above 0.6 in thirteen climate zones except BSh and BWh because of the sparse distribution of training stations. We suggest researchers who work on SSM predictions with ML to use the single ML algorithms RFR and KNR, and the ensemble
models KNR_RFR_XB.

In summary, our results showed that there is a huge potential to use ensemble models to generate accurate SSM products globally which is important to local-scale environment and agricultural applications.

Code and data availability. The algorithms in this paper were conducted in Python. The code is available inhttps://doi.org/10.5281/zenodo.8004346 (Han et al. 2023a). The training data is available on request to Qianqian Han (q.han@utwente.nl).

**Author Contributions (CRediT author statement):** Yijian Zeng, Bob Su, Qianqian Han, Lijie Zhang, Calimanut-Iount
Cira, and Brigitta Szabo conceptualised and designed this study. Qianqian Han, Lijie Zhang, Calimanut-Iount Cira, Egor Prikaziuk, Brigitta Szabo, and Ting Duan wrote the codes and did the analysis. Qianqian Han, Lijie Zhang, Chao Wang



wrote the original draft. Yijian Zeng, Bob Su, Calimanut-Icount Cira, and Brigitta Szabo, Chao Wang, and Salvatore Manfreda provided guidance and technical inputs to this study. All authors participated in the discussions and provided guidance and advice throughout the experimental design and data validation process, and all reviewed the manuscript. All
authors have read and agreed to the published version of the manuscript.

**Data Availability Statement:** The data used in this study are publicly available but subjected to registration and ISMN data policy as the maintainer of the original data (the International Soil Moisture Network, or ISMN) stated in its "Terms and Conditions" section that "No onward distribution: Re-export or transfer of the original data (as received from the ISMN archive) by the data users to a third party is prohibited." (ISMN, 2023).

**Acknowledgment:** The research presented in this paper was funded in part by the China Scholarship Council (grant no.202004910427). The authors would like to thank Virtual Mobility Grant from HARMONIOUS project. We are grateful for the freely available data at GEE, and the in-situ data from ISMN.

**Conflicts of Interest:** The authors declare no conflict of interest. The funders had no role in the design of the study; in the collection, analyses, or interpretation of data; in the writing of the manuscript; or in the decision to publish the results.

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
