# Peer review of "Ensemble of optimised machine learning algorithms for predicting surface soil moisture content at global scale"

_Geoscientific Model Development, 2023_

## Referee Comment (RC1)

This paper presents an interesting and comprehensive study on the use of machine learning (ML) algorithms for predicting surface soil moisture content at a global scale. The authors use eight different machine learning algorithms trained with in situ data to predict soil moisture at the top 5 cm of the soil across the globe at a spatial resolution of 1 km and a temporal coverage of 2000-2018. The performance of various ML algorithms was evaluated. They also created 10 ensemble models from five optimized base models with the highest performance metrics and propose KNR_RFR_XB as the best ensemble for soil moisture prediction.

The paper provides a thorough description of the model building and validation process. However, prior to publication, the following questions should be addressed:

**Abstract**

Please specify which soil layer, period (coverage), temporal and spatial resolution were considered in the generation of the machine learning models.

Line 15-20: Instead of using the term "best data product", it would be more appropriate to use an adjective such as reliable or well-validated, as the data product from machine learning may be produced empirically without explicit knowledge of the physical processes involved. This may introduce additional uncertainties into the final output. (O., S., Orth, R. Global soil moisture data derived through machine learning trained with in-situ measurements. Sci Data 8, 170 (2021). https://doi.org/10.1038/s41597-021-00964-1 )

Line 24-34: Please clarify what it means to have the best performance on "test_random" set, "test_temporal" and "test_independent-stations" and reword this sentence to focus on the meaning rather than the jargon terms. Also, please use full wording before notation (XB = extreme gradient boosting).

**Introduction**

Please provide citations for all claims or statements that require them (e. g., Line 42:45, Line 47:49, Line 52:54)

Line 65-68: The statement "Another advantage of using ML techniques is that they can help to reduce the uncertainty…" may not be accurate, as machine learning may introduce additional uncertainties into the final output (see response to line 15:20). It would be helpful to discuss this issue in more detail and provide references to support your argument.

Line 75: Please specify which soil layer, period (coverage), temporal and spatial resolution were considered in the generation of the machine learning models.

**Data**

Line 91:92: Did you filter out NAN values from the predictor data based on the in-situ soil moisture data? If not, what did you do? Please make the statement clear.

Line 104: Why is "Year/ DOY " a predictor?

Line 140: To improve readability, could you please ensure that all temporal coverages are consistent (2000-2018). I noticed that line 117 mentions the years 2000-2019, but I believe only the years 2000-2018 were used for analysis.

Line 232: " The full data contained a total of 735,475 registries (each sample contained nineteen predictor variables) and was differentiated in the training set and three different test sets, based on the following strategy (also described in Table 2)"

I am finding this a bit confusing and would appreciate some clarification. Is it correct that the combined data of all predictors and in-situ data is 735,475, and that this data was split into train and test samples as shown in Table 2?

Additionally, I noticed that Table 1 shows fewer than 19 predictors. Could you please explain how you arrived at the number 19 predictors? Is it necessary to use all 19 predictors, or could feature or predictor importance be performed to determine the various influences of predictors and select important features if necessary?

**Methodology**

Line 279: Introduce the full wording e.g., coefficient of determination before using the associated notation $r^2$.

Line 350: Could you please explain the reason for choosing both $r$ and $r^2$ as evaluation metrics? What is the significance of using both in this context?

Line 384: Arranging Table 3, as well as any other relevant tables, in ascending or descending order based on performance can improve readability.

Line 430: 470: Could you please explain why the Kruskal-Wallis test is necessary when you can already compare models using various performance metrics such as RMSE and $r^2$ ?

Line 480: Please move Figure 5 into supplementary information.

In addition to addressing these questions, it would also be helpful to compare the performance metrics and uncertainties of your proposed ensemble model with other existing soil moisture products or methods. This would provide more context for your results and help readers evaluate their significance. It would also be interesting to discuss the limitations and assumptions of your machine learning algorithms and how they affect the reliability and applicability of your soil moisture predictions. Finally, it would be useful to suggest future work or research directions to improve or extend your study.

---

## Author Comment (AC6)

**Reply to Reviewer #1**

**General comments**

*This paper presents an interesting and comprehensive study on the use of machine learning (ML) algorithms for predicting surface soil moisture content at a global scale. The authors use eight different machine learning algorithms trained with in situ data to predict soil moisture at the top 5 cm of the soil across the globe at a spatial resolution of 1 km and a temporal coverage of 2000-2018. The performance of various ML algorithms was evaluated. They also created 10 ensemble models from five optimized base models with the highest performance metrics and propose KNR_RFR_XB as the best ensemble for soil moisture prediction.*

*The paper provides a thorough description of the model building and validation process. However, prior to publication, the following questions should be addressed:*

Thanks for your thorough review and detailed comments on our manuscript. Your comments are immensely valuable in enhancing the manuscript's quality.

**Specific comments**
**Abstract**

*Please specify which soil layer, period (coverage), temporal and spatial resolution were considered in the generation of the machine learning models.*

*1. Line 15-20: Instead of using the term "best data product", it would be more appropriate to use an adjective such as reliable or well-validated, as the data product from machine learning may be produced empirically without explicit knowledge of the physical processes involved. This may introduce additional uncertainties into the final output. (O., S., Orth, R. Global soil moisture data derived through machine learning trained with in-situ measurements. Sci Data 8, 170 (2021). https://doi.org/10.1038/s41597-021- 00964-1 )*

Reply 1: Thanks for your advice. We agree with your comment. In line 18, we will change "the best" into "reliable".

*2. Line 24-34: Please clarify what it means to have the best performance on "test_random" set, "test_temporal" and "test_independent-stations" and reword this sentence to focus on the meaning rather than the jargon terms. Also, please use full wording before notation (XB = extreme gradient boosting).*

Reply 2: Thanks for your advice. As what we understood, here we need to clarify two things: (1) meaning of "best performance", (2) explain a bit of these 3 terms: "test_random" set, "test_temporal" and "test_independent-stations".

In line 23-24, we will modify the sentence "The result showed that K-neighbours Regressor (KNR) performs best on "test_random" set, while Random Forest Regressor (RFR) performs best on "test_temporal" and "test_independent-stations." to a more accurate description with quantitative indicators: "The result showed that K-neighbours Regressor (KNR) had the lowest Root Mean Square Error (0.0379 cm$^3$/cm$^3$) on "test_random" set, while Random Forest Regressor (RFR) had the lowest RMSE (0.0599 cm$^3$/cm$^3$) on "test_temporal" and AdaBoost (AB) had the lowest RMSE (0.0786 cm$^3$/cm$^3$) on "test_independent-stations".

We will explain the meaning of "test_random": for testing the performance of randomly split data during training, "test_temporal": for testing the performance on the period which were not used in training, "test_independent-stations": for testing the performance on the stations which were not used in training.

In line 33, we will replace "XB" with "Extreme gradient Boosting".

**Introduction**

*3. Please provide citations for all claims or statements that require them (e. g., Line 42:45, Line 47:49, Line 52:54)*

Reply 3: Thank you very much for your valuable suggestion. We will ensure that all the claims and statements, which you've highlighted as requiring citations, will have proper citations in our revised manuscript. These citations will contribute to the accuracy and reliability of the paper.

*4. Line 65-68: The statement "Another advantage of using ML techniques is that they can help to reduce the uncertainty…" may not be accurate, as machine learning may introduce additional uncertainties into the final output (see response to line 15:20). It would be helpful to discuss this issue in more detail and provide references to support your argument.*

Reply 4: Thanks for your advice. In our study, when we said ML can help reduce the uncertainty, we initially meant that compared to traditional physical models, ML can alleviate uncertainties (e.g. on parameter optimization) to some extent. Nowadays people integrate ML and physical models to reduce the prediction uncertainty (Roy et al. 2023). However, we agree that machine learning itself has uncertainties. To provide a more accurate representation of our standpoint, we will remove this statement and will state the limitations of machine learning in the subsequent discussion section which is related to the 16th comment.

*5. Line 75: Please specify which soil layer, period (coverage), temporal and spatial resolution were considered in the generation of the machine learning models.*

Reply 5: Thanks for your attentive review. In line 78, we will add the information: for predicting SSM in 0-5 cm depth with daily and 1 km resolution from 2000 to 2018.

**Data**

*6. Line 91:92: Did you filter out NAN values from the predictor data based on the in-situ soil moisture data? If not, what did you do? Please make the statement clear.*

Reply 6: Thanks for your advice and sorry if this was unclear in the manuscript. We first organized the in-situ soil moisture data, and then extracted the predictor variables for each station in the period when they have in-situ soil moisture data. After this, we filtered out NAN values for each data pair (soil moisture and all predictor variables in one specific time and location). If there are NAN values in either soil moisture or all predictor variables, we removed it. (Because we organized the in-situ soil moisture data in a specific period (2000-2018), there is missing soil moisture values during this period as well).

To make this statement clear, in lines 92-93, we will change from "In this study, we extracted in-situ SSM from ISMN from 1 January 2000 to 31 December 2018, filtered the NaN (Not a Number) values of different predictor variables and kept 1722 stations for further analysis (Figure 1). Land surface information was also incorporated to prepare training and testing sets." to "In this study, we extracted in-situ SSM from ISMN from 1 January 2000 to 31 December 2018, and then the predictor variables were incorporated to prepare training and testing sets. We filtered the NaN (Not a Number) values of SSM and different predictor variables, and after this 1722 stations were kept for further analysis (Figure 1)."

*7. Line 104: Why is "Year/ DOY " a predictor?*

Reply 7: Thank you pointing out this question. There are two reasons why we include Year and DOY as predictors.

(1) Firstly, they can help capture the seasonal variations in soil moisture. Soil moisture can exhibit significant changes across different seasons, such as summer droughts or winter wet periods. By incorporating year and DOY as input variables, the model can learn and capture these seasonal trends, leading to more accurate predictions of soil moisture changes at different time points.

(2) Secondly, Year and DOY help analyze the historical trends of soil moisture. Certain years or specific dates might exhibit consistent trends in soil moisture, such as prolonged dry spells or continuous rainfall. The model can leverage this historical trend information to make more accurate predictions of soil moisture changes.

In addition, we will implement the feature importance experiment (which is related to the 10th comment) to provide more details.

*8. Line 140: To improve readability, could you please ensure that all temporal coverages are consistent (2000-2018). I noticed that line 117 mentions the years 2000-2019, but I believe only the years 2000-2018 were used for analysis.*

Reply 8: Thanks for your careful check. The correct temporal coverate should be 2000-2018. In line 117, "2019" will be changed to "2018".

*9. Line 232: " The full data contained a total of 735,475 registries (each sample contained nineteen predictor variables) and was differentiated in the training set and three different test sets, based on the following strategy (also described in Table 2)" I am finding this a bit confusing and would appreciate some clarification. Is it correct that the combined data of all predictors and in-situ data is 735,475, and that this data was split into train and test samples as shown in Table 2?*

Reply 9 : Yes, that is correct. We aimed to explain how we split the full data into 4 parts. Our full data has 735475 rows, and each row includes in-situ soil moisture and 19 predictor variables.

*10. Additionally, I noticed that Table 1 shows fewer than 19 predictors. Could you please explain how you arrived at the number 19 predictors? Is it necessary to use all 19 predictors, or could feature or predictor importance be performed to determine the various influences of predictors and select important features if necessary?*

Reply 10: Thanks for your careful check. Because in Table1 we wrote "Soil Texture", actually it includes sand, silt, and clay content. In the fifth row and the first column of Table1, we will modify "Soil Texture" to "Clay Content/Sand Content/Silt Content". In the fifth row and the second column, we will modify "Soil texture (proportion of clay, sand, silt)" to "Proportion of clay/proportion of sand/proportion of silt".

For the feature importance, thank you very much for your valuable feedback. In response to your question regarding the necessity of using such a multitude of input variables, we will conduct a comprehensive feature importance analysis across different regression models to address this inquiry. The KNR+RFR+XB ensemble model performed the best, therefore we plan to compute the feature importance based on these three MLs, and then we calculate the mean feature importance of all features for these three MLs with permutation importance method (Li et al. 2021).

We believe by doing this experiment, we will observe the feature importance across different ML models and the overall feature importance. Then we can conclude that which input variables contribute to predicting the target variable. The result of this experiment will be added in the supplementary information.

**Methodology**

*11. Line 279: Introduce the full wording e.g., coefficient of determination before using the associated notation r 2.*

Reply 11: Thanks for your careful check. In line 279, we will add the full wording of $r^2$, from "the $r^2$ score" to "the Coefficient of Determination R-square ($r^2$) score".

*12. Line 350: Could you please explain the reason for choosing both r and r 2 as evaluation metrics? What is the significance of using both in this context?*

Reply 12: Thank you for your advice. The r score measures the correlation between two or more variables (A and B, for example). The $r^2$ score measures how much variation of B values can be explained by A values). They are two very different metrics and each measures a different thing. The $r^2$ score is more important in ML training. Based on your advice, we decided to keep $r^2$ score as evaluation metric to provide a comprehensive view of the model's performance, accounting for its ability to explain variance.

However, in the evaluation of individual independent sites (section 4.4 and 4.5), we observed instances where r² values could be negative due to poorer model performance on some sites because they were not included in the training and different climate conditions from training sites. These situations might potentially lead to misinterpretations of model performance. To ensure an accurate assessment, we will keep using r score values in this evaluation part, which directly quantify the linear relationship between predicted and observed values.

Based on above, section 4.4 and 4.5 which are related to individual independent stations evaluation we will keep same as before. Other parts we will remove the r score values.

*13. Line 384: Arranging Table 3, as well as any other relevant tables, in ascending or descending order based on performance can improve readability.*

Reply 13: That is a nice idea, sorting will be done based on increasing RMSE. Table 3 will be changed into sorting on increasing RMSE. Table 4 and 5 will be changed into sorting on increasing RMSE on "Test_random".

*14. Line 430: 470: Could you please explain why the Kruskal-Wallis test is necessary when you can already compare models using various performance metrics such as RMSE and r 2 ?*

Reply 14: We use the Kruskal-Wallis test to analyze if there is significant difference between the performance of the different MLs and ensemble models based on the squared errors of the predictions, therefore we find it important to show the results of this test. We see the metrics but those are single values and statistical tests are the way of distribution comparison.

*15. Line 480: Please move Figure 5 into supplementary information.*

Reply 15: Thanks for your advice. We agree with it. Figure 5 will be moved into supplementary information.

*16. In addition to addressing these questions, it would also be helpful to compare the performance metrics and uncertainties of your proposed ensemble model with other existing soil moisture products or methods. This would provide more context for your results and help readers evaluate their significance. It would also be interesting to discuss the limitations and assumptions of your machine learning algorithms and how they affect the reliability and applicability of your soil moisture predictions. Finally, it would be useful to suggest future work or research directions to improve or extend your study.*

Reply 16: Thanks for your valuable advice. We will modify like following:

(1) In line 596, we will add the comparison with other methods and soil moisture products. We drafted an initial paragraph could be added: "Furthermore, a comparison of ensemble models with other existing

soil moisture products and methods highlights their superior performance. For instance, evaluating the root mean square error (RMSE) on randomly selected test samples, our KNR_RFR_XB ensemble model achieves an RMSE of 0.0355 cm³/cm³. In contrast, the RFR model used for generating a global daily 1 km soil moisture product presents an ubRMSE (unbiased root-mean-square error) of 0.045 cm³/cm³ (Zheng et al. 2023) and the RFR model used for generating a global daily 0.25 degree soil moisture product presents an RMSE of 0.05 cm³/cm³ (Zhang et al. 2021). The RFR model used for reconstruction of a daily SMAP surface soil moisture dataset shows an ubRMSE of 0.04 cm³/cm³ (Yang and Wang 2023). The XB model used for generating a global daily 1 km soil moisture product presents an RMSE of 0.038 cm³/cm³ (Zhang et al. 2023). These comparisons demonstrate that our ensemble model has the potential to improve predictive accuracy compared to individual methods. This makes our model a candidate for further exploration as an effective tool for accurate prediction of soil moisture."

 (2) For the uncertainty, limitations and future work. We will rewrite the content in lines 618-627, and move the content in lines 597-604 to lines 628-. We will add the future work in the end of this part. The final version of the uncertainty, limitations and future work will be like: "Although the proposed ensemble model has been demonstrated to be an effective solution to predict soil moisture, there are still limitations. Firstly, our ensemble model can be limited in the areas outside the training conditions  such as climate zones BSh, BWh and BWk (Sungmin and Orth 2021).  Secondly, hyper-parameters tuning is a computationally expensive operation that proved to have an important effect on the performance of each machine learning model. However, it involves the human factor with expertise in choosing the right ranges for each hyperparameter in order to achieve the best possible training. We recommend to carry out the training in at least two iterations, first selecting wider parameter intervals, and then narrowing it down to ranges in proximity to the best value detected in the initial experiments. Thirdly, the training of algorithmic implementations within ensembling environments requires more computational power. However, the increased and the more stable prediction behaviour is more desirable when tackling tasks where high performance metrics are expected. Lastly, depending on the base models, the performance of ensemble models can sometimes worsen when compared to well-performing optimised algorithms. For this reason, it is advised to optimise the base algorithms as much as possible for the chosen task. It is also observed that regression algorithms with a higher complexity generally displayed a higher generalisation capacity. The above four points highlight the limitations and challenges of our ensemble model in practical applications. Future research directions may include enhancing the generalization ability of the model to obtain more accurate predictions in areas outside the training conditions, such as increasing the training data or using transfer learning techniques. In addition, more efficient and automated hyperparameter tuning methods can be explored to improve the performance of the model. Addressing the computational demands required to train the ensemble models is also a key direction, possibly involving the use of parallel computing, distributed frameworks, or hardware acceleration approaches, all of which aim to further enhance the performance and applicability of our models."

The content we will add above is also related to the 4[th] comment.

**References**

Li, W., Migliavacca, M., Forkel, M., Walther, S., Reichstein, M., & Orth, R. Revisiting global vegetation controls using multi‑layer soil moisture. *Geophysical Research Letters, 48*, e2021GL092856, 2021

Roy, A., Kasiviswanathan, K., Patidar, S., Adeloye, A.J., Soundharajan, B.S., & Ojha, C.S.P. A physics‑aware machine learning‑based framework for minimizing the prediction uncertainty of hydrological models. *Water Resour. Res.*, e2023WR034630, 2023

Sungmin, O., & Orth, R. Global soil moisture data derived through machine learning trained with in-situ measurements. *Sci. Data, 8*, 1-14, doi:10.1038/s41597-021-00964-1*, 2021

Yang, H., & Wang, Q. Reconstruction of a spatially seamless, daily SMAP (SSD_SMAP) surface soil moisture dataset from 2015 to 2021. *J. Hydrol., 621*, 129579, 2023

Zhang, L., Zeng, Y., Zhuang, R., Szabó, B., Manfreda, S., Han, Q., & Su, Z. In Situ Observation-Constrained Global Surface Soil Moisture Using Random Forest Model. *Remote Sens., 13*, 4893, doi:10.3390/rs13234893*, 2021

Zhang, Y., Liang, S., Ma, H., He, T., Wang, Q., Li, B., Xu, J., Zhang, G., Liu, X., & Xiong, C. Generation of global 1 km daily soil moisture product from 2000 to 2020 using ensemble learning. *Earth Syst. Sci. Data, 15*, 2055-2079, 10.5194/essd-15-2055-2023*, 2023

Zheng, C., Jia, L., & Zhao, T. A 21-year dataset (2000–2020) of gap-free global daily surface soil moisture at 1-km grid resolution. *Sci. Data, 10*, 139, 2023

---

## Author Comment (AC7)

**Reply to Reviewer #2:**

**General comments**

Based on global-scale data, this study considered the use of integrated machine learning models to investigate the estimation of daily soil moisture (SSM). The results of the study demonstrate that the integrated machine learning algorithm outperforms both the optimization and base machine learning algorithms in predicting SSM. The topic of SSM estimation is of significant importance. I have the following suggestions and questions to further enhance the current manuscript:

Thanks for the kind words! Your comments are immensely valuable in enhancing the manuscript's quality.

**Specific comments**

*1. I would suggest including a literature review on data selection in the introduction, while reserving the Data section solely for describing the data used and the data preprocessing.*

Reply 1: Thank you for bringing up this important point! The selection of features is an integral part of the ML methodology (relevance of the features/variables, length of data, structure of ML algorithms and their tuning), the data/features selection is based on previous works by others and our own insights (will be demonstrated in the feature importance figures). In this case, we will reorganize the section 2, by renaming "2. Data" to "2. Physical Features and Data". The secondary title will be:

- 2.1 Physical Features Selection
- 2.2 Data Source
- 2.3 Data Pre-processing
- 2.4 Data Split.

Then we will move the feature selection part from lines 109-112, 120-124, 131-138, 143-155, 163-171 to section "2.1 Physical Features Selection". In the beginning of section 2.1, we will add: "In order to predict SSM accurately, a multidimensional understanding of its complex dynamics requires a comprehensive integration of diverse environmental factors. While remote sensing techniques and advanced machine learning algorithms have revolutionized SSM estimation, the optimal selection of predictor variables remains a pivotal challenge. The dynamic interplay of precipitation, evaporation, land surface temperature (LST), vegetation index, soil properties, and topographic indices influences SSM patterns." to get an overview before explaining every predictor variable.

In the beginning of section 2.2, we will mention "Based on consideration of these physical features that influence that dynamics of SSM, we next describe the data used for training and testing the different algorithms."

*2. This study investigates the utilization of integrated machine learning models. However, it raises the question of whether three base models in the integrated model are optimal. Have the authors considered the possibility of adjusting the number of models, such as exploring whether a model integrated with two machine learning models may yield better performance?*

Reply 2: Thanks for your great suggestion. We built the ensemble models from three MLs to derive more robust and stable predictions than that of a single algorithm. It was an experimental design decision taken so that each ensemble structure is based on the same number of base models. The optimization of the number of the models included in the ensemble model is out of the scope of this manuscript but an interesting topic for a future study, thank you for highlighting it.

*3. Page 3, Line 80: "(ii)justify ... model". I would suggest adding experimental results or analyses in this area, such as manipulating the effect of a predictor on the results by increasing or decreasing its influence.*

Reply 3: Thanks for your advice. In response to the comment regarding the justification of the model choice (ii) on page 3, line 80, we will further expand our analysis to provide additional insights into the rationale behind our selection. We plan to conduct a comprehensive feature importance analysis across different regression models, aiming to assess the impact of each predictor on the model outcomes. Specifically, the KNR+RFR+XB ensemble model performed the best, therefore we plan to compute the feature importance based on these three MLs, and then we calculate the mean feature importance of all features for these three MLs with permutation importance method (Li et al. 2021).

We believe by doing this experiment, we will observe the feature importance across different ML models and the overall feature importance. Then we can conclude that which input variables contribute to predicting the target variable. The result of this experiment will be added in the supplementary information.

*4. Page 12, Line 255-271. The authors chose eight machine learning models based on their popularity and performance, but it appears that there is a significant amount of research applying artificial neural networks (ANN) or using them as a baseline (Uthayakumar et al., 2022; Senyurek et al., 2020; Liu et al., 2020...). Surprisingly, the authors did not include ANN in their selection.*

Reply 4: Thank you for your perceptive observation and thoughtful suggestion regarding our choice of machine learning models. In our study, we selected eight machine learning models based on their established popularity and documented performance within the literature. While it is true that artificial neural networks (ANN) have garnered significant attention and application in various studies, we did incorporate the most simple ANN, namely the multilayer perceptron regressor (MLPR), into our study, in order to identify future lines of research. We appreciate your insights and the incorporation of additional ANN algorithms is an interesting topic for a future research.

*5. The manuscript is too long and needs to be reduced, and the authors of the figures and tables need to be revised and optimized.*

Reply 5: Thanks for your advice. In the revised manuscript, we already moved table 2 to supplementary information, it was changed to table S1. We also moved Figure 4 into supplementary information, it was changed to Figure S4. The r score column/columns were removed from Table 3,4,5, also decreased their size. In the end, all tables have been reformatted to increase readability.

**References**

Li, W., Migliavacca, M., Forkel, M., Walther, S., Reichstein, M., & Orth, R. Revisiting global vegetation controls using multi‐layer soil moisture. *Geophysical Research Letters, 48*, e2021GL092856, 2021

---

## Author Response (AR1)

**Reply to Reviewer #1**

**General comments**

This paper presents an interesting and comprehensive study on the use of machine learning (ML) algorithms for predicting surface soil moisture content at a global scale. The authors use eight different machine learning algorithms trained with in situ data to predict soil moisture at the top 5 cm of the soil across the globe at a spatial resolution of 1 km and a temporal coverage of 2000-2018. The performance of various ML algorithms was evaluated. They also created 10 ensemble models from five optimized base models with the highest performance metrics and propose KNR_RFR_XB as the best ensemble for soil moisture prediction.

The paper provides a thorough description of the model building and validation process. However, prior to publication, the following questions should be addressed:

Thanks for your thorough review and detailed comments on our manuscript. Your comments are immensely valuable in enhancing the manuscript's quality.

**Specific comments**

**Abstract**

Please specify which soil layer, period (coverage), temporal and spatial resolution were considered in the generation of the machine learning models.

*1. Line 15-20: Instead of using the term "best data product", it would be more appropriate to use an adjective such as reliable or well-validated, as the data product from machine learning may be produced empirically without explicit knowledge of the physical processes involved. This may introduce additional uncertainties into the final output. (O., S., Orth, R. Global soil moisture data derived through machine learning trained with in-situ measurements. Sci Data 8, 170 (2021). https://doi.org/10.1038/s41597-021- 00964-1 )*

Reply 1: Thanks for your advice. We agree with your comment. In line 18-19, we changed "the best" into "reliable".

*2. Line 24-34: Please clarify what it means to have the best performance on "test_random" set, "test_temporal" and "test_independent-stations" and reword this sentence to focus on the meaning rather than the jargon terms. Also, please use full wording before notation (XB = extreme gradient boosting).*

Reply 2: Thanks for your advice. As what we understood, here we need to clarify two things: (1) meaning of "best performance", (2) explain a bit of these 3 terms: "test_random" set, "test_temporal" and "test_independent-stations". In line 23-28, we changed from "The result showed that K-neighbours Regressor (KNR) performs best on "test_random" set, while Random Forest Regressor (RFR) performs best on "test_temporal" and "test_independent-stations." to "The result showed that K-neighbours Regressor (KNR) had the lowest Root Mean Square Error (0.0379 cm$^3$/cm$^3$) on "test_random" set (for testing the performance of randomly split data during training), while Random Forest Regressor (RFR) had the lowest RMSE (0.0599 cm$^3$/cm$^3$) on "test_temporal" set (for testing the performance on the period which were not used in training) and AdaBoost (AB) had the lowest RMSE (0.0786 cm$^3$/cm$^3$) on "test_independent-stations" set (for testing the performance on the stations which were not used in training).

In line 36, we also replaced "XB" with "Extreme gradient Boosting".

**Introduction**

*3. Please provide citations for all claims or statements that require them (e. g., Line 42:45, Line 47:49, Line 52:54)*

Reply 3: Thanks for your advice.

- In lines 46-50 we added 2 citations.
    - The first citation was added in line 47 (Zhang et al. 2022) and the reference is in lines 964-966: Zhang, P., Zheng, D., van der Velde, R., Wen, J., Ma, Y., Zeng, Y., Wang, X., Wang, Z., Chen, J., & Su, Z. A dataset of 10-year regional-scale soil moisture and soil temperature measurements at multiple depths on the Tibetan Plateau. Earth Syst. Sci. Data, 14, 5513-5542, 2022.
    - The second citation was added in line 50 (Srivastava et al. 2016) and the reference is in line 926: Srivastava, P.K., Petropoulos, G.P., & Kerr, Y.H. Satellite soil moisture retrieval: techniques and applications. Elsevier, 2016.
- In line 52-54 we added 2 citations.
    - The first citation was added in line 53 (Njoku and Entekhabi 1996) and the reference is in lines 890: Njoku, E.G., & Entekhabi, D. Passive microwave remote sensing of soil moisture. J. Hydrol., 184, 101-129, 1996.
    - The second citation was added in line 54 (Al Bitar et al. 2017) and the reference is in lines 788-789: Al Bitar, A., Mialon, A., Kerr, Y.H., Cabot, F., Richaume, P., Jacquette, E., Quesney, A., Mahmoodi, A., Tarot, S., & Parrens, M. The global SMOS Level 3 daily soil moisture and brightness temperature maps. Earth Syst. Sci. Data, 9, 293-315, 2017.
- In lines 58-60 we added 1 citation
    - The citation was added in line 60 (Piles et al. 2011) and the reference is in lines 903-904: Piles, M., Camps, A., Vall-Llossera, M., Corbella, I., Panciera, R., Rudiger, C., Kerr, Y.H., & Walker, J. Downscaling SMOS-derived soil moisture using MODIS visible/infrared data. IEEE T. Geosci. Remote, 49, 3156-3166, 2011.

*4. Line 65-68: The statement "Another advantage of using ML techniques is that they can help to reduce the uncertainty…" may not be accurate, as machine learning may introduce additional uncertainties into the final output (see response to line 15:20). It would be helpful to discuss this issue in more detail and provide references to support your argument.*

Reply 4: Thanks for your advice. In our study, when we said ML can help reduce the uncertainty, we initially mean that compared to traditional physical models, ML can alleviate uncertainties (e.g. on parameter optimization) to some extent. Nowadays people integrate ML and physical models to reduce the prediction uncertainty (Roy et al. 2023). However, we agree that machine learning itself has uncertainties. To provide a more accurate representation of our standpoint, we removed this statement and will state the limitations of machine learning in the subsequent discussion section which is related to the 16th comment.

*5. Line 75: Please specify which soil layer, period (coverage), temporal and spatial resolution were considered in the generation of the machine learning models.*

Reply 5: Thanks for your advice. In lines 81-87, we changed from "Here we aim to optimise the prediction of soil moisture with training data distributed across the globe by ensemble models constructed from different base ML algorithms and extensively study their performances in order to identify optimized combinations for predicting soil moisture." to "Here we aim to optimise the prediction of SSM with training data distributed across the globe by ensemble models constructed from different base ML algorithms and extensively study their performances in order to identify optimized combinations for predicting SSM at 5 cm depth. It is note that current predicted SSM

product is at point scale with daily temporal resolution, and based on data availability of ISMN stations, we are predicting SSM from 2000 to 2018. The developed model can be easily and directly upscaled to predict SSM at global scale with 1km resolution if the input data is provided (Han et al. 2023; Zhang et al. 2021), which will be produced in the future and is beyond the scope of the current study."

**Data**

*6. Line 91:92: Did you filter out NAN values from the predictor data based on the in-situ soil moisture data? If not, what did you do? Please make the statement clear.*

Reply 6: Thanks for your advice. We first organized the in-situ soil moisture data, and then extracted the predictor variables for each station in the period when they have in-situ soil moisture data. After this, we filtered out NAN values for each data pair (soil moisture and all predictor variables in one specific time and location). If there are NAN values in either soil moisture or all predictor variables, we removed it. (Because we organized the in-situ soil moisture data in a specific period (2000-2018), there were missing soil moisture values during this period as well).

To make this statement clear, in lines 167-169, we changed from "In this study, we extracted in-situ SSM from ISMN from 1 January 2000 to 31 December 2018, filtered the NaN (Not a Number) values of different predictor variables and kept 1722 stations for further analysis (Figure 1). Land surface information was also incorporated to prepare training and testing sets." to "In this study, we extracted in-situ SSM (at 5cm depth) from ISMN from 1 January 2000 to 31 December 2018, and then the predictor variables were incorporated to prepare training and testing sets. We filtered the NaN (Not a Number) values of SSM and different predictor variables, and after this 1722 stations were kept for further analysis (Figure 1)."

*7. Line 104: Why is "Year/ DOY " a predictor?*

Reply 7: Thank you pointing out this question. There are two reasons why we include Year and DOY as predictors.

(1) Firstly, they can help capture the seasonal variations in soil moisture. Soil moisture can exhibit significant changes across different seasons, such as summer droughts or winter wet periods. By incorporating year and DOY as input variables, the model can learn and capture these seasonal trends, leading to more accurate predictions of soil moisture changes at different time points.

(2) Secondly, Year and DOY help analyze the historical trends of soil moisture. Certain years or specific dates might exhibit consistent trends in soil moisture, such as prolonged dry spells or continuous rainfall. The model can leverage this historical trend information to make more accurate predictions of soil moisture changes.

In addition, the feature importance result (see the reply to comment 10) also proved that Year and DOY play crucial roles in soil moisture prediction.

*8. Line 140: To improve readability, could you please ensure that all temporal coverages are consistent (2000-2018). I noticed that line 117 mentions the years 2000-2019, but I believe only the years 2000- 2018 were used for analysis.*

Reply 8: Thanks for your careful check. It should be 2000-2018. In line 191, "2019" was changed to "2018".

*9. Line 232: " The full data contained a total of 735,475 registries (each sample contained nineteen predictor variables) and was differentiated in the training set and three different test sets, based on the following strategy (also described in Table 2)" I am finding this a bit confusing and would appreciate*

*some clarification. Is it correct that the combined data of all predictors and in-situ data is 735,475, and that this data was split into train and test samples as shown in Table 2?*

Reply 9: Yes, that is correct. We aimed to explain how we split the full data into 4 parts. Our full data has 735475 rows, and each row includes in-situ soil moisture and 19 predictor variables.

*10. Additionally, I noticed that Table 1 shows fewer than 19 predictors. Could you please explain how you arrived at the number 19 predictors? Is it necessary to use all 19 predictors, or could feature or predictor importance be performed to determine the various influences of predictors and select important features if necessary?*

Reply 10: Thanks for your careful check. Because in Table1 we wrote "Soil Texture", actually it includes sand, silt, and clay content. In line 181, in the fifth row and the first column of Table1, we modified "Soil Texture" to "Clay Content/Sand Content/Silt Content". In the fifth row and the second column was also changed from "Soil texture (proportion of clay, sand, silt)" to "Proportion of clay/proportion of sand/proportion of silt".

For the feature importance, thank you very much for your valuable feedback. In response to your question regarding the necessity of using such a multitude of input variables, we have conducted a comprehensive feature importance analysis across different regression models to address this inquiry. The KNR+RFR+XB ensemble model performed the best, therefore we compute the feature importance based on these three MLs, and then we calculated the mean feature importance of all features for these three MLs with permutation importance method (Li et al. 2021).

We observed variations in feature importance across different ML models. Each model highlights distinct features, indicating diverse contributions of these variables in different models. By aggregating the feature importance results from the three ML models, we concluded that all 19 input variables collectively contribute to predicting the target variable. Notably, features such as API, lon, and DOY are consistently influential across multiple models. Interestingly, our analysis suggests that the variable precipitation demonstrates relatively lower importance across these three ML models on predicting soil moisture. This intriguing revelation prompts the consideration of future research endeavors aimed at unraveling the nuanced relationship between precipitation and the target variable. As such investigation is beyond the scope of the current work, we defer the exploration of machine learning models with a reduced number of predictors to subsequent studies. Implementing such an approach would necessitate significant adjustments in experimental design and research objectives, accompanied by a substantial increase in computational resources to re-run all training experiments. The supplementary information lines 17-45 provides further details on these results.

**Methodology**

*11. Line 279: Introduce the full wording e.g., coefficient of determination before using the associated notation r 2.*

Reply 11: Thanks for your careful check. In lines 356-357, we added the full wording of $r^2$, from "the $r^2$ score" to "the Coefficient of Determination R-square ($r^2$) score".

*12. Line 350: Could you please explain the reason for choosing both r and r 2 as evaluation metrics? What is the significance of using both in this context?*

Reply 12: Thank you for your advice. The r score measures the correlation between two or more variables (A and B, for example). The $r^2$ score measures how much variation of B values can be explained by A values). They are two very different metrics and each measures a different thing. The $r^2$ score is more important in ML training. Based on your advice, we decided to keep $r^2$ score as evaluation metric to provide a comprehensive view of the model's performance, accounting for its ability to explain variance.

However, in the evaluation of individual independent sites (section 4.4 and 4.5), we observed instances where r² values could be negative due to poorer model performance on some sites because they were not included in the training and have different climate conditions from training sites. These situations might potentially lead to misinterpretations of model performance. To ensure an accurate assessment, we opted to use r score values in this evaluation part, which directly quantify the linear relationship between predicted and observed values.

Based on above, section 4.4 and 4.5 which are related to individual independent stations evaluation we keep the same as before. Other parts we modified are below:

In line 436, we deleted "the r score" because we removed r score from the second step.

In line 463, we removed the r score column from table3. In line 478, we removed three r score columns from table 4. In line 506, we removed three r score columns from table5.

In line 484 and 485, we changed from "Specifically, KNR_RFR_XB and GB_RFR_XB displayed the best performance in "test_random" (RMSE were 0.0355 $cm^3/cm^3$ and 0.0391 $cm^3/cm^3$, and the r scores were 0.9488 and 0.9379) and test_temporal (RMSE were 0.0576 $cm^3/cm^3$ and 0.0568 $cm^3/cm^3$ and r scores were 0.8571 and 0.8614) sets." to "Specifically, KNR_RFR_XB and GB_RFR_XB displayed the best performance in "test_random" (RMSE were 0.0355 $cm^3/cm^3$ and 0.0391 $cm^3/cm^3$, and the $r^2$ scores were 0.8985 and0.8772) and test_temporal (RMSE were 0.0576 $cm^3/cm^3$ and 0.0568 $cm^3/cm^3$ and $r^2$ scores were 0.7335 and 0.7410) sets."

In line 496 we added "and a $r^2$ score value of 0.8985". In line 497-499, we changed from "and the r scores of KNR, RFR, and XB were 0.9407, 0.9301, and 0.9385. Compared with KNR, RFR, and XB, the ensemble model improved, for RFR, 0.0058 cm3/cm3 (14%) of RMSE and 0.0187 (2%) of r score. " to "and the $r^2$ scores of KNR, RFR, and XB were 0.8848, 0.8626, and0.8806. Compared with KNR, RFR, and XB, the ensemble model improved, for RFR, 0.0058 cm3/cm3 (14%) of RMSE and 0.0137 (1.6%) of $r^2$ score."

In lines 545-552, we changed r score into $r^2$ score in 6 subfigures of Figure 4, and this Figure 4 was moved to supplementary information based on the 4th comment from the second reviewer, it is "Figure S4" now.

In line 642, we changed from "a high r score of 0.9407" to "a high $r^2$ score of 0.8848".

In line 643, we changed from "highest r score of 0.8446" to "highest $r^2$ score of 0.7126".

In line 644, we changed from "highest r score of 0.7005" to "highest $r^2$ score of 0.4905".

*13. Line 384: Arranging Table 3, as well as any other relevant tables, in ascending or descending order based on performance can improve readability.*

Reply 13: Good idea, sorting is based on increasing RMSE. Table 3 was changed into sort on increasing RMSE. Table 4 and 5 were changed into sort on increasing RMSE on "test_random".

*14. Line 430: 470: Could you please explain why the Kruskal-Wallis test is necessary when you can already compare models using various performance metrics such as RMSE and r 2 ?*

Reply 14: We use the Kruskal-Wallis test to analyze if there is significant difference between the performance of the different MLs and ensemble models based on the squared errors of the predictions, therefore we find it important to show the results of this test. We see the metrics but those are single values and statistical tests are the way of distribution comparison.

*15. Line 480: Please move Figure 5 into supplementary information.*

Reply 15: Thanks for your advice. We agree with it. Figure 5 was changed into Figure S2. In line 565, we changed "Figure 5" into "Figure S2". The original "Figure S1" was changed into "Figure S3" in the supplementary information.

*16. In addition to addressing these questions, it would also be helpful to compare the performance metrics and uncertainties of your proposed ensemble model with other existing soil moisture products or methods. This would provide more context for your results and help readers evaluate their significance. It would also be interesting to discuss the limitations and assumptions of your machine learning algorithms and how they affect the reliability and applicability of your soil moisture predictions. Finally, it would be useful to suggest future work or research directions to improve or extend your study.*

Reply 16: Thanks for your advice. We modified as following:

(1) In lines 678-688, we added the comparison with other methods and soil moisture products.

"Furthermore, a comparison of ensemble models with other existing soil moisture products and methods highlights their superior performance. For instance, evaluating the root mean square error (RMSE) on randomly selected test samples, our KNR_RFR_XB ensemble model achieves an RMSE of 0.0355 cm³/cm³. In contrast, the RFR model used for generating a global daily 1 km soil moisture product presents an ubRMSE (unbiased root-mean-square error) of 0.045 cm³/cm³ (Zheng et al. 2023) and the RFR model used for generating a global daily 0.25 degree soil moisture product presents an RMSE of 0.05 cm³/cm³ (Zhang et al. 2021). The RFR model used for reconstruction of a daily SMAP surface soil moisture dataset shows an ubRMSE of 0.04 cm³/cm³ (Yang and Wang 2023). The XB model used for generating a global daily 1 km soil moisture product presents an RMSE of 0.038 cm³/cm³ (Zhang et al. 2023). These comparisons demonstrate that our ensemble model has the potential to improve predictive accuracy compared to individual methods. This makes our model a candidate for further exploration as an effective tool for accurate prediction of soil moisture."

(2) For the uncertainty, limitations and future work. We rewrote the content in lines 710-720, and moved the content in lines 689-696 to lines 721727. We added the future work in lines 727-733. The final version of the uncertainty, limitations and future work is in lines 710-733: "Although the proposed ensemble model has been demonstrated to be an effective solution to predict soil moisture, there are still limitations. Firstly, our ensemble model can be limited in the areas outside the training conditions such as climate zones BSh, BWh and BWk (Sungmin and Orth 2021). Secondly, hyper-parameters tuning is a computationally expensive operation that proved to have an important effect on the performance of each machine learning model. However, it involves the human factor with expertise in choosing the right ranges for each hyperparameter in order to achieve the best possible training. We recommend to carry out the training in at least two iterations, first selecting wider parameter intervals, and then narrowing it down to ranges in proximity to the best value detected in the initial experiments. Thirdly, the training of algorithmic implementations within ensembling environments requires more computational power. However, the increased and the more stable prediction behaviour is more desirable when tackling tasks where high performance metrics are expected. Lastly, depending on the base models, the performance of ensemble models can sometimes worsen when compared to well-performing optimised algorithms. For this reason, it is advised to optimise the base algorithms as much as possible for the chosen task. It is also observed that regression algorithms with a higher complexity generally displayed a higher generalisation capacity. The above four points highlight the limitations and challenges of our ensemble model in practical applications. Future research directions may include enhancing the generalization ability of the model to obtain more accurate predictions in areas outside the training conditions, such as increasing the training data or using transfer learning techniques. In addition, more efficient and automated hyperparameter tuning methods can be explored to improve the performance of the model. Addressing the computational demands required to train the ensemble models is also a key direction, possibly

involving the use of parallel computing, distributed frameworks, or hardware acceleration approaches, all of which aim to further enhance the performance and applicability of our models."

The content we added above is also related to the 4th comment.

*17. Additional revision*

Reply 17: In lines 56-57, we changed from "special sensor microwave/imager (SSM/I), soil moisture and ocean salinity (SMOS)" to "Special Sensor Microwave/Imager (SSM/I), Advanced Microwave Scanning Radiometer for Earth Observation system (AMSR-E), Soil Moisture and Ocean Salinity (SMOS)"

In lines 75-79: we changed from "At point scale, Uthayakumar compared three ML algorithms in the laboratory using a Radar Sensor (Uthayakumar et al. 2022). On a regional scale, Adab Acharya and Senyurek compared different ML approaches over catchment areas or at larger regional scales (Acharya et al. 2021; Adab et al. 2020; Senyurek et al. 2020). Liu compared six ML algorithms in generating high-resolution SSM over four regions (Liu et al. 2020)" to "At point scale, study compared three ML algorithms in the laboratory using a Radar Sensor (Uthayakumar et al. 2022). On a regional scale, studiescompared different ML approaches over catchment areas or at larger regional scales (Acharya et al. 2021; Adab et al. 2020; Senyurek et al. 2020) and six ML algorithms was compared in generating high-resolution SSM over four regions (Liu et al. 2020)"

In line 330, we renamed "3. Methodology" to "3. ML Algorithms and Ensembles" because other parts that belong to methodology are discussed in section 2 and 4 also.

In line 171-172, to make it more clear, we changed from "It is worth noting that the original ISMN data is not fully openly accessible, and registration is required for further inquiry or downloading. It is to note that the original ISMN data is accessible after registration (ISMN, 2023)" to "It is worth noting that registration is required for further inquiry or downloading the original ISMN data (ISMN, 2023). "

**Reply to Reviewer #2**

**General comments**

Based on global-scale data, this study considered the use of integrated machine learning models to investigate the estimation of daily soil moisture (SSM). The results of the study demonstrate that the integrated machine learning algorithm outperforms both the optimization and base machine learning algorithms in predicting SSM. The topic of SSM estimation is of significant importance. I have the following suggestions and questions to further enhance the current manuscript:

**Specific comments**

*1. I would suggest including a literature review on data selection in the introduction, while reserving the Data section solely for describing the data used and the data preprocessing.*

Reply 1: Thank you for bringing up this important point! The selection of features is an integral part of the ML methodology (relevance of the features/variables, length of data, structure of ML algorithms and their tuning), the data/features selection is based on previous works by others and our own insights (as demonstrated in the feature importance figures). In this case, we will reorganize the section 2, by renaming "2. Data" to "2. Physical Features and Data". The section "2.1 Land Surface Features" will be split into two sections "2.1 Physical Features Selection" and "2.2 Data Source". The secondary title will be changed from:

- 2.1 Land Surface Features
- 2.2 Data Pre-processing
- 2.3 Data Split

to the following:

- 2.1 Physical Features Selection
- 2.2 Data Source
- 2.3 Data Pre-processing
- 2.4 Data Split.

The numbers of the corresponding third-level headings will also be modified: from 2.1.1-6 to 2.2.1-6, 2.2.1-6 to 2.3.1-6.

Then we will move the feature selection part from lines 183-186, 194-198, 205-212, 217-229, 237-245 to section "2.1 Physical Features Selection" in lines 93-136. In the beginning of section 2.1 (lines 94-98), we added: "In order to predict SSM accurately, a multidimensional understanding of its complex dynamics requires a comprehensive integration of diverse environmental factors. While remote sensing techniques and advanced machine learning algorithms have revolutionized SSM estimation, the optimal selection of predictor variables remains a pivotal challenge. The dynamic interplay of precipitation, evaporation, land surface temperature (LST), vegetation index, soil properties, and topographic indices influences SSM patterns." to get an overview before explaining every predictor variable.

In the beginning of section 2.2 (lines 158-159), we mentioned "Based on consideration of these physical features that influence that dynamics of SSM, we next describe the data used for training and testing the different algorithms."

Lines 137-156 will be moved to section 2.2 in lines 159-177.

In line 246, we added "MERIT Hydro was used in this study (Yamazaki et al. 2019)."

*2. This study investigates the utilization of integrated machine learning models. However, it raises the question of whether three base models in the integrated model are optimal. Have the authors considered the possibility of adjusting the number of models, such as exploring whether a model integrated with two machine learning models may yield better performance?*

Reply 2: Thanks for your great suggestion. We built the ensemble models from three MLs to derive more robust and stable predictions than that of a single algorithm. It was an experimental design decision taken so that each ensemble structure is based on the same number of base models. The optimization of the number of the models included in the ensemble model is out of the scope of this manuscript but an interesting topic for a future study, thank you for highlighting it.

*3. Page 3, Line 80: "(ii)justify … model". I would suggest adding experimental results or analyses in this area, such as manipulating the effect of a predictor on the results by increasing or decreasing its influence.*

Reply 3: Thanks for your advice. In response to the comment regarding the justification of the model choice (ii) on page 3, line 86, we have further expanded our analysis to provide additional insights into the rationale behind our selection. We conducted a comprehensive feature importance analysis across different regression models, aiming to assess the impact of each predictor on the model outcomes. Specifically, the KNR+RFR+XB ensemble model performed best, therefore we computed the feature importance based on these three MLs, and then we calculated the mean feature importance of all features for these three MLs with permutation importance method (Li et al. 2021). This comment is related to the 10[th] comment from the first reviewer. Please find more details in the supplementary information lines 17-45.

*4. Page 12, Line 255-271. The authors chose eight machine learning models based on their popularity and performance, but it appears that there is a significant amount of research applying artificial neural networks (ANN) or using them as a baseline (Uthayakumar et al., 2022; Senyurek et al., 2020; Liu et al., 2020...). Surprisingly, the authors did not include ANN in their selection.*

Reply 4: Thank you for your perceptive observation and thoughtful suggestion regarding our choice of machine learning models. In our study, we selected eight machine learning models based on their established popularity and documented performance within the literature. While it is true that artificial neural networks (ANN) have garnered significant attention and application in various studies, we did incorporate the most simple ANN, namely the multilayer perceptron regressor (MLPR), into our study, in order to identify future lines of research. We appreciate your insights and the incorporation of additional ANN algorithms is an interesting topic for a future research.

*5. The manuscript is too long and needs to be reduced, and the authors of the figures and tables need to be revised and optimized.*

Reply 5: Thanks for your advice. We moved table 2 to supplementary information, it was changed to table S1. In line 444 and line 457, "Table 2" was changed to "Table S1". We also moved Figure 4 to supplementary information, it was changed to Figure S4. In line 540, "Figure 4" was changed to "Figure S4". The r score column/columns were removed from Table 3,4,5, also decreased their size. In the end, all tables have been reformatted to increase readability.

**References**

Acharya, U., Daigh, A.L., & Oduor, P.G. Machine Learning for Predicting Field Soil Moisture Using Soil, Crop, and Nearby Weather Station Data in the Red River Valley of the North. *Soil Systems, 5*, 57, doi:10.3390/soilsystems5040057*, 2021*

Adab, H., Morbidelli, R., Saltalippi, C., Moradian, M., & Ghalhari, G.A.F. Machine learning to estimate surface soil moisture from remote sensing data. *Water, 12*, 3223, doi:10.3390/w12113223*, 2020*

Han, Q., Zeng, Y., Zhang, L., Wang, C., Prikaziuk, E., Niu, Z., & Su, B. Global long term daily 1 km surface soil moisture dataset with physics informed machine learning. *Sci. Data, 10*, 101, doi:10.1038/s41597-023-02011-7*, 2023*

Li, W., Migliavacca, M., Forkel, M., Walther, S., Reichstein, M., & Orth, R. Revisiting global vegetation controls using multi‐layer soil moisture. *Geophysical Research Letters, 48*, e2021GL092856, 2021

Liu, Y., Jing, W., Wang, Q., & Xia, X. Generating high-resolution daily soil moisture by using spatial downscaling techniques: A comparison of six machine learning algorithms. *Adv. Water Resour., 141*, 103601, doi:10.1016/j.advwatres.2020.103601*, 2020*

Roy, A., Kasiviswanathan, K., Patidar, S., Adeloye, A.J., Soundharajan, B.S., & Ojha, C.S.P. A physics‐aware machine learning‐based framework for minimizing the prediction uncertainty of hydrological models. *Water Resour. Res.*, e2023WR034630, 2023

Senyurek, V., Lei, F., Boyd, D., Kurum, M., Gurbuz, A.C., & Moorhead, R. Machine learning-based CYGNSS soil moisture estimates over ISMN sites in CONUS. *Remote Sens., 12*, 1168, doi:10.3390/rs12071168*, 2020*

Sungmin, O., & Orth, R. Global soil moisture data derived through machine learning trained with in-situ measurements. *Sci. Data, 8*, 1-14, doi:10.1038/s41597-021-00964-1*, 2021*

Uthayakumar, A., Mohan, M.P., Khoo, E.H., Jimeno, J., Siyal, M.Y., & Karim, M.F. Machine learning models for enhanced estimation of soil moisture using wideband radar sensor. *Sensors, 22*, 5810, doi:10.3390/s22155810*, 2022*

Yamazaki, D., Ikeshima, D., Sosa, J., Bates, P.D., Allen, G.H., & Pavelsky, T.M. MERIT Hydro: A high‐resolution global hydrography map based on latest topography dataset. *Water Resour. Res., 55*, 5053-5073, doi:10.1029/2019WR024873*, 2019*

Yang, H., & Wang, Q. Reconstruction of a spatially seamless, daily SMAP (SSD_SMAP) surface soil moisture dataset from 2015 to 2021. *J. Hydrol., 621*, 129579, 2023

Zhang, L., Zeng, Y., Zhuang, R., Szabó, B., Manfreda, S., Han, Q., & Su, Z. In Situ Observation-Constrained Global Surface Soil Moisture Using Random Forest Model. *Remote Sens., 13*, 4893, doi:10.3390/rs13234893*, 2021*

Zhang, Y., Liang, S., Ma, H., He, T., Wang, Q., Li, B., Xu, J., Zhang, G., Liu, X., & Xiong, C. Generation of global 1 km daily soil moisture product from 2000 to 2020 using ensemble learning. *Earth Syst. Sci. Data, 15*, 2055-2079, 10.5194/essd-15-2055-2023*, 2023*

Zheng, C., Jia, L., & Zhao, T. A 21-year dataset (2000–2020) of gap-free global daily surface soil moisture at 1-km grid resolution. *Sci. Data, 10*, 139, 2023

---

## Author Response (AR2)

1. Firstly, we changed the order of 3 figures.

1) In lines 460,461,462,465,466,473, we changed Figure 6 to Figure 4.

2) In line 503, we changed Figure 7 to Figure 5.

3)In lines 495 and 505, we changed Figure 8 to Figure 6.

2. Secondly, we compiled the figures that have multiple panels into one figure, that is Figure 4, Figure 6, Figure S1, Figure S3, Figure S4. We also changed these 5 figures with the new version in our manuscript and supplementary information.

3. Thirdly, in line 636-644, we changed from "**Acknowledgment:** The research presented in this paper was funded in part by the China Scholarship Council (grant no.202004910427). The authors would like to thank Virtual Mobility Grant from HARMONIOUS project. We are grateful for the freely available data at GEE, and the in-situ data from ISMN." to "**Acknowledgment:** The research presented in this paper was funded in part by the China Scholarship Council (grant no.202004910427). The authors would like to thank Virtual Mobility Grant from HARMONIOUS project. This research has been funded by The Netherlands Organisation for Scientific Research (NWO) KIC, WUNDER project (grant no. KICH1. LWV02.20.004), Netherlands eScience Center, EcoExtreML project (grant ID. 525 27020G07) and Water JPI project "iAqueduct" (Project number: ENWWW.2018.5). In addition, this study was supported in part by the ESA ELBARA-II/III Loan Agreement EOP-SM/2895/TC-tc and the ESA MOST Dragon IV Program. We also thank the National Natural Science Foundation of China (grant no. 41971033), Fundamental Research Funds for the Central Universities, CHD (grant no. 300102298307), MIUR PON R&I 2014-2020 Program (project MITIGO, ARS01_00964). We are grateful for the freely available data at GEE, and the in-situ data from ISMN."